# The zinc-finger transcription factor Sfp1 imprints specific classes of mRNAs and links their synthesis to cytoplasmic decay

Moran Kelbert[1†], Antonio Jordán-Pla[2†], Lola de Miguel-Jiménez[3†], José García-Martínez[2], Michael Selitrennik[1], Adi Guterman[1], Noa Henig[1], Sander Granneman[4], José E Pérez-Ortín[2*], Sebastián Chávez[3*], Mordechai Choder[1*]

[1]Department of Molecular Microbiology, Rappaport Faculty of Medicine, Technion-Israel Institute of Technology, Haifa, Israel; [2]Instituto Biotecmed, Facultad de Biológicas, Universitat de València, Burjassot, Spain; [3]Instituto de Biomedicina de Sevilla, Universidad de Sevilla-CSIC-Hospital Universitario Virgen del Rocío, and Departamento de Genética, Facultad de Biología, Universidad de Sevilla, Seville, Spain; [4]Centre for Engineering Biology, School of Biological Sciences, University of Edinburgh, Edinburgh, United Kingdom

*For correspondence:
jose.e.perez@uv.es (JEP-O);
schavez@us.es (SC);
choder@technion.ac.il (MC)

†These authors contributed equally to this work

Competing interest: The authors declare that no competing interests exist.

**Abstract** To function effectively as an integrated system, the transcriptional and post-transcriptional machineries must communicate through mechanisms that are still poorly understood. Here, we focus on the zinc-finger Sfp1, known to regulate transcription of proliferation-related genes. We show that Sfp1 can regulate transcription either by binding to promoters, like most known transcription activators, or by binding to the transcribed regions (gene bodies), probably via RNA polymerase II (Pol II). We further studied the first mode of Sfp1 activity and found that, following promoter binding, Sfp1 binds to gene bodies and affects Pol II configuration, manifested by dissociation or conformational change of its Rpb4 subunit and increased backtracking. Surprisingly, Sfp1 binds to a subset of mRNAs co-transcriptionally and stabilizes them. The interaction between Sfp1 and its client mRNAs is controlled by their respective promoters and coincides with Sfp1's dissociation from chromatin. Intriguingly, Sfp1 dissociation from the chromatin correlates with the extent of the backtracked Pol II. We propose that, following promoter recruitment, Sfp1 accompanies Pol II and regulates backtracking. The backtracked Pol II is more compatible with Sfp1's relocation to the nascent transcripts, whereupon Sfp1 accompanies these mRNAs to the cytoplasm and regulates their stability. Thus, Sfp1's co-transcriptional binding imprints the mRNA fate, serving as a paradigm for the cross-talk between the synthesis and decay of specific mRNAs, and a paradigm for the dual-role of some zinc-finger proteins. The interplay between Sfp1's two modes of transcription regulation remains to be examined.

## eLife assessment

This **important** study reports that a transcription factor stimulating mRNA synthesis can stabilize its target transcripts. The **convincing** results demonstrate, with multiple independent approaches, co-transcriptional binding, stabilization of a family of mRNAs, and cytoplasmic activities of the transcription factor Sfp1. The results lead to the conclusion that the co-transcriptional association of Sfp1 with specific transcripts is a critical step in the stabilization of such transcripts in the cytoplasm.

**eLife digest** The ability to fine-tune the production of proteins in a cell is essential for organisms to exist. An imbalance in protein levels can be the cause of various diseases. Messenger RNA molecules (mRNA) link the genetic information encoded in DNA and the produced proteins. Exactly how much protein is made mostly depends on the amount of mRNA in the cell's cytoplasm. This is controlled by two processes: the synthesis of mRNA (also known as transcription) and mRNA being actively degraded.

Although much is known about mechanisms regulating transcription and degradation, how cells detect if they need to degrade mRNA based on the levels of its synthesis and vice versa is poorly understood. In 2013, researchers found that proteins known as 'RNA decay factors' responsible for mRNA degradation are actively moved from the cell's cytoplasm into its nucleus to instruct the transcription machinery to produce more mRNA.

Kelbert, Jordán-Pla, de-Miguel-Jiménez et al. – including some of the researchers involved in the 2013 work – investigated how mRNA synthesis and degradation are coordinated to ensure a proper mRNA level. The researchers used advanced genome engineering methods to carefully manipulate and measure mRNA production and degradation in yeast cells.

The experiments revealed that the protein Sfp1 – a well-characterized transcription factor for stimulating the synthesis of a specific class of mRNAs inside the nucleus – can also prevent the degradation of these mRNAs outside the nucleus. During transcription, Sfp1 bound directly to mRNA. The investigators could manipulate the co-transcriptional binding of Sfp1 to a certain mRNA, thereby changing the mRNA stability in the cytoplasm.

This suggests that the ability of Sfp1 to regulate both the production and decay of mRNA is dependent on one another and that transcription can influence the fate of its transcripts. This combined activity can rapidly change mRNA levels in response to changes in the cell's environment.

RNA plays a key role in ensuring correct levels of proteins. It can also function as an RNA molecule, independently of its coding capacity. Many cancers and developmental disorders are known to be caused by faulty interactions between transcription factors and nucleic acids. The finding that some transcription factors can directly regulate both mRNA synthesis and its destruction introduces new angles for studying and understanding these diseases.

## Introduction

In the last decade, investigators have accumulated evidence for a direct cross-talk between the transcriptional and post-transcriptional stages of gene expression. The dialogue between transcription and mRNA decay has been reported by several investigators, including us (*Begley et al., 2019*; *Blasco-Moreno et al., 2019*; *Bregman et al., 2011*; *Bryll and Peterson, 2023*; *Buratti and Baralle, 2012*; *Chattopadhyay et al., 2022*; *El-Brolosy et al., 2019*; *Gilbertson et al., 2018*; *Goler-Baron et al., 2008*; *Haimovich et al., 2013b*; *Harel-Sharvit et al., 2010*; *Hartenian and Glaunsinger, 2019*; *Ma et al., 2019*; *Sun et al., 2012*; *Timmers and Tora, 2018*; *Trcek et al., 2011*; *Vera et al., 2014*; *Zid and O'Shea, 2014*). Often, this dialog results in 'mRNA buffering,' which maintains an approximately constant mRNA level despite transient changes in mRNA synthesis or decay rates (*Haimovich et al., 2013b*; *Sun et al., 2012*). In particular, communication pathways have been discovered between promoters and mRNA translation (*Chen et al., 2022*; *Vera et al., 2014*; *Zid and O'Shea, 2014*) or decay (*Bregman et al., 2011*; *Trcek et al., 2011*). However, the underlying mechanism remains elusive.

During transcription by Pol II, several RNA-binding proteins (RBPs) bind to nascent transcripts, thus modulating the transcription cycle (*Battaglia et al., 2017*). Importantly, in some cases, RBPs remain bound to the mRNA, accompanying it to the cytoplasm and regulating post-transcriptional stages. For example, the Pol II subunits Rpb4 and Rpb7 bind to mRNAs and regulate their translation and decay (*Goler-Baron et al., 2008*; *Harel-Sharvit et al., 2010*). We named the latter cases 'mRNA imprinting,' because this binding imprints the mRNA fate (*Choder, 2011*; *Dahan and Choder, 2013*). Although Rpb4 is a quintessential imprinted factor, it functions as a general factor that binds to numerous mRNAs (*Garrido-Godino et al., 2021*). Imprinting by a 'classical' transcription factor (TF) that binds to specific promoters, thus regulating transcription by modulating the assembly of general transcription

factors, including Pol II, is currently unknown. Such imprinting might broaden our understanding of the functions of TFs – not just transcription regulators but regulators of both the nuclear transcription and cytoplasmic post-transcription stages of gene expression.

We previously reported a special case of cross-talk between promoter elements and mRNA decay which is mediated by Rap1 binding sites (RapBSs) and by Rap1 protein (*Bregman et al., 2011*). RapBSs (12–14 bp) are found in ~5% of the promoters, including ~90% of ribosomal protein (RP) and a small portion of Ribosome Biogenesis (RiBi) promoters (*Lascaris et al., 1999*; *Lieb et al., 2001*; *Shore et al., 2021*; *Warner, 1999*). Rap1 is bound to essentially all such RP promoters in vivo (*Lieb et al., 2001*; *Schawalder et al., 2004*) and is involved in their transcription activation. Rap1 functions as a pioneering transcription factor (TF) required for the binding of a pair of RP-specific TFs called Fhl1 and Ifh1 (*Lieb et al., 2001*; *Shore et al., 2021*). An additional TF recruited by Rap1 is Sfp1 (see next paragraph). Previously, we demonstrated that Rap1 plays a dual role in maintaining the level of specific mRNAs – stimulating both mRNA synthesis and decay. We proposed that Rap1 represents a class of factors, synthegradases, whose recruitment to promoters stimulates (or represses) both mRNA synthesis and decay (*Bregman et al., 2011*). How Rap1 stimulates mRNA decay is unknown.

The split-finger protein 1 (Sfp1) is a zinc-finger TF, the deletion or overexpression of which has been shown to affect the recruitment of the TATA-binding protein (TBP) and Pol II to RiBi and RiBi-like genes, and to affect the transcription of RP and snoRNA genes (*Albert et al., 2019*; *Jorgensen et al., 2004*; *Marion et al., 2004*; *Fingerman et al., 2003*). It also impacts many G1/S ('START') genes, where it appears to act as a repressor (*Albert et al., 2019*). Sfp1 was, therefore, viewed as a classical TF. Shore's lab has examined the Sfp1 configuration within promoters, using either chromatin immuno-precipitation (ChIP-seq) or chromatin endogenous cleavage (ChEC-seq), and found that the two approaches exhibit different sensitivity to different configurations. While Sfp1 binding sites detected by ChEC are enriched for the motif gAAAATTTTc, binding identified by ChIP is dependent on Ifh1, hence on Rap1 as well (*Albert et al., 2019*). Thus, Sfp1 can be viewed as a 'classical' zinc-finger containing TF. Zinc-finger domains have also been described to function as RNA-binding domains (*Font and MacKay, 2010*). In some proteins, such as TFIIIA, the same zinc-finger domain can recognize both a specific double-stranded DNA sequence and another specific sequence in single-stranded RNA - discussed in *Font and MacKay, 2010*.

In response to the misincorporation of nucleotides, damage to the DNA, nucleosomes, hairpins or certain DNA sequences, elongating Pol II transiently pauses. During pausing, Pol II remains cata-lytically active and can resume transcription if the obstacle is removed (*Bar-Nahum et al., 2005*; *Gómez-Herreros et al., 2012a*). However, these pauses increase the probability of Pol II backward movement along the DNA template, a process named 'backtracking.' During backtracking, the 3' end of the nascent RNA extrudes from the active site, traps the trigger loop of the enzyme, and stably blocks mRNA elongation (*Cheung and Cramer, 2011*). Once arrested, Pol II can reactivate itself by cleaving its own synthetized RNA at the active site (*Sigurdsson et al., 2010*). Cleavage is substantially enhanced by TFIIS (*Churchman and Weissman, 2011*), which becomes essential under NTP scarcity (*Archambault et al., 1992*). Important for this paper, Pol II backtracking is highly frequent at RP and RiBi genes, compared to other gene regulons, and is highly dependent on Rap1 (*Pelechano et al., 2009*). Our previous work has also shown that several RP regulators, including Rap1 and Sfp1, make transcription of RP genes highly dependent on TFIIS, especially under conditions of transcriptional stress due to NTP depletion (*Gómez-Herreros et al., 2012b*). Thus, Sfp1 can be considered as a backtracking regulator. Regulators that antagonize Pol II backtracking are Xrn1 (*Begley et al., 2021*; *Begley et al., 2019*; *Fischer et al., 2020*), Rpb4 (*Fischer et al., 2020*) and the-Ccr4-Not complex (*Collart, 2016*), which prevents backtracking in a manner that depends on its Rpb4 and Rpb7 subunits (*Babbarwal et al., 2014*; *Kruk et al., 2011*). Interestingly, backtracking and TFIIS recruitment in RP genes are not influenced by Xrn1, but are strongly affected by Ccr4 (*Begley et al., 2019*).

Here, we assigned a much broader function to a TF in gene expression. We discovered that Sfp1 regulates both transcription and mRNA decay through its capacity to bind RNAs co-transcription-ally. The Sfp1-mediated cross-talk between mRNA synthesis and decay results in the cooperation of these opposing processes to increase mRNA abundance, by stimulating the former and repressing the latter. This cooperation is limited to a subgroup of Sfp1 targets, most of which are also Rap1 targets. Our results also unveil a role for some promoters as mediators between RPB and its interacting RNAs.

## Results

### Sfp1 interacts with Rpb4

Previously, we discovered that the yeast Pol II subunits, Rpb4 and Rpb7, which form a heterodimer Rpb4/7, have post-transcriptional roles. Rpb4/7 is recruited onto mRNAs co-transcriptionally and is directly involved in all major post-transcriptional stages of the mRNA lifecycle such as RNA export (*Farago et al., 2003*) translation (*Harel-Sharvit et al., 2010*) and decay (*Goler-Baron et al., 2008*; *Lotan et al., 2007*; *Lotan et al., 2005*). Thus, nascent mRNA emerges from the nucleus with 'imprinted' information that serves to regulate post-transcriptional stages of gene expression (*Dahan and Choder, 2013*). To identify novel factors that may be involved in 'mRNA imprinting,' we searched for Rpb4-interacting proteins, using a yeast two-hybrid screen. Among several interacting partners, *SFP1* was singled out, as it is a known transcription factor of genes encoding RPs. RP genes represent a subclass of genes known to be preferentially regulated by Rpb4 at the level of mRNA degradation (*Lotan et al., 2005*). The interaction of Rpb4 with Sfp1 was corroborated by pairwise two-hybrid analysis and co-immunoprecipitation experiment (*Figure 1—figure supplement 1A* and results not shown), as well as by an imaging approach (see later). Note that no interaction was observed with Fhl1, Ifh1, and Abf1, which encode additional transcription factors that often function in concert with Sfp1 (see Introduction). It is important to note that our two hybrid, immunoprecipitation and imaging results do not differentiate between direct or indirect interactions.

### Transcription-dependent export of Sfp1 from the nucleus to the cytoplasm

As a first step in determining whether Sfp1 plays a post-transcriptional role outside the nucleus, we examined its ability to shuttle between the nucleus and cytoplasm. GFP-Sfp1 was expressed in WT cells and in cells carrying the temperature-sensitive (ts) allele of *NUP49* (*nup49-313*), which encodes a nucleoporin used in a standard shuttling assay (*Selitrennik et al., 2006* references therein). Following a temperature shift up to 37 °C to inactivate Nup49-dependent import, GFP-Sfp1 accumulated in the cytoplasm (*Figure 1A*, '*nup49-313*'), indicating that this protein normally shuttles between the nucleus and cytoplasm, at least at 37 °C. We next simultaneously disabled both protein import and transcription, and by using cells with a double ts mutation in both *NUP49* and in the Pol II subunit gene, *RPB1* (referred to as *nup49-313(ts), rpb1-1* double mutant cells). Simultaneous temperature inactivation of both import and transcription, using these double mutant cells, abolished Sfp1-GFP export (*Figure 1A*, compare '*nup49-313*' and '*nup49-313, rpb1-1*'). This suggests that Sfp1 export to the cytoplasm is transcription-dependent, similar to Rpb4 (*Selitrennik et al., 2006*). Previously, it was observed that Sfp1 accumulates in the cytoplasm of cells when they are starved (*Marion et al., 2004*). In our experiment, we provided glucose to starved cells and monitored the import of GFP-Sfp1 in both normal (WT) cells and those lacking the Rpb4 protein (*rpb4Δ*). We found that the import of GFP-Sfp1 was slower in cells lacking Rpb4, which supports the idea that there is a connection between Sfp1 and Rpb4 (*Figure 1—figure supplement 1B, C*).

### Sfp1 is localized to P-bodies

During starvation, Sfp1 localizes to discrete cytoplasmic foci. These foci represent P-bodies - phase-separated droplets containing RNAs, mRNA decay factors, and other proteins (*Luo et al., 2018*) - as they contain the P-body markers Dcp2 and Lsm1, as well as Rpb4 (*Figure 1B*), and decrease in number if cells are pre-treated with cycloheximide before starvation (*Luo et al., 2018*; *Figure 1C*).

### Sfp1p binds to a specific set of mRNAs

The observation that Sfp1 is exported in a transcription-dependent manner (*Figure 1B*) raised the possibility that Sfp1 is exported in association with mRNAs, similar to Rpb4 (*Duek et al., 2018*; *Goler-Baron et al., 2008*). To examine this possibility, we performed a UV cross-linking and analysis of cDNA (CRAC) (*Granneman et al., 2009*). This approach is both highly reliable and can map the position within the RNA where the UV crosslinking occurs at nearly nucleotide resolution (*Bohnsack et al., 2012*; *Haag et al., 2017*). The endogenous *SFP1* gene was surgically tagged with an N-terminal His6-TEV-Protein A (HTP) tag, retaining intact 5' and 3' non-coding regions. Disruption of *SFP1* results in slow growth and a small cell size (*Jorgensen et al., 2004*). N-terminal tagging permitted normal

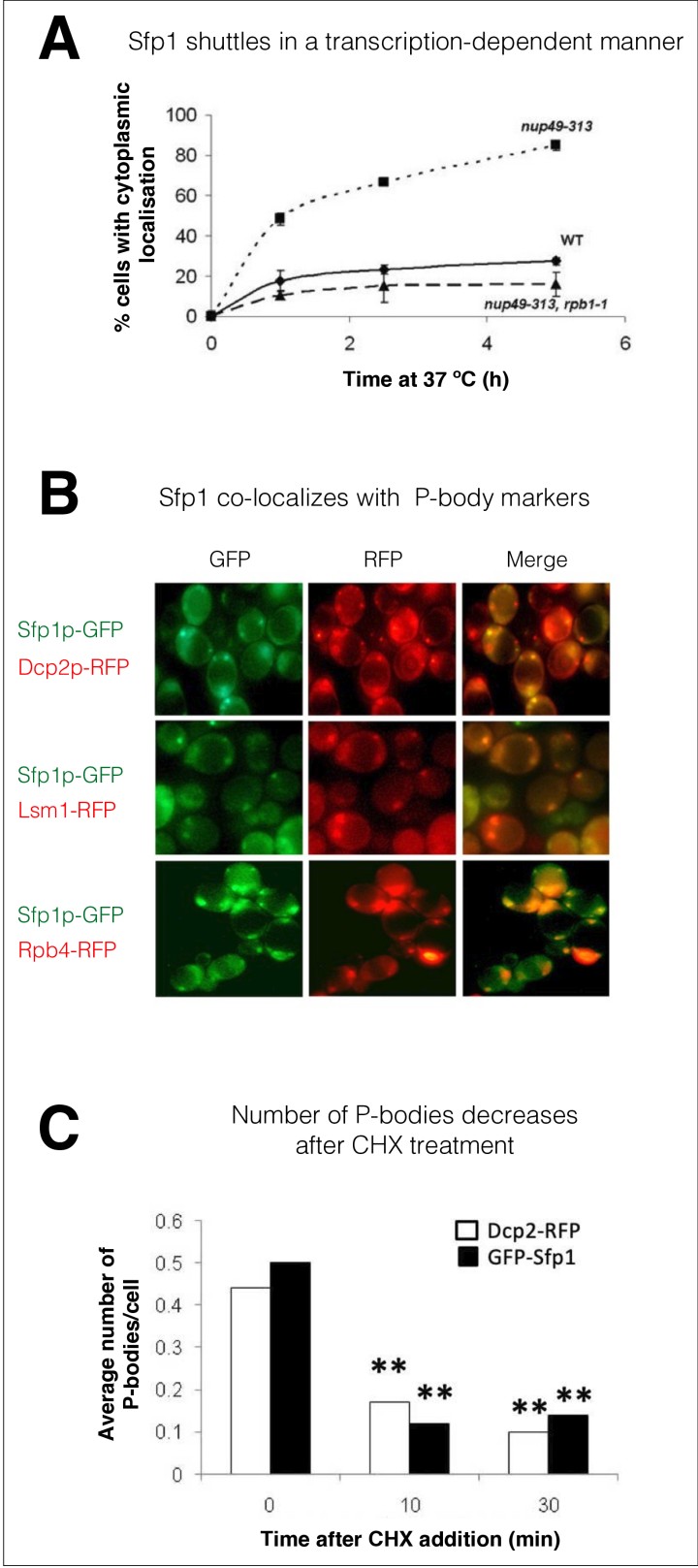

**Figure 1.** Split-finger protein 1 (Sfp1) shuttles in a transcription-dependent manner and localizes to P-bodies. (**A**) *Shuttling assay*. The assay used the temperature-sensitive (ts) *nup49-313* mutant and was performed as reported (***Lee et al., 1996***; ***Selitrennik et al., 2006***). Wild-type (WT, yMS119), *nup49-313*(ts) (yMS1), and *nup49-313*(ts) *rpb1-1*(ts) cells expressing GFP-Sfp1 were allowed to proliferate under optimal conditions at 24 °C. Cycloheximide

*Figure 1 continued on next page*

*Figure 1 continued*

(CHX) (50 µg/ml) was then added, and the cultures were shifted to 37 °C (to inhibit Nup49-313 and Rpb1-1). The proportion of cells expressing cytoplasmic GFP-Sfp1 was plotted as a function of time (n>200). Error bars represent the standard deviation of three replicates. (**B**) *GFP-Sfp1 is co-localized with P-bodies markers.* Cells expressing the indicated fluorescent proteins were allowed to proliferate till the mid-logarithmic phase, followed by 24- hr starvation in a medium lacking glucose and amino acids. Live cells were inspected under the fluorescent microscope. White arrows mark P-bodies. (**C**) *The number of GFP-Sfp1-containing foci per cell decreases in response to cycloheximide (CHX) treatment.* CHX (50 µg/ml) was added to exponentially proliferating cultures for the indicated time. Cells were then shifted to starvation medium as in B; Average of two replicates is shown. Student's t-test between time 0 and the indicated time points was performed; **p<0.001 (n≥240).

The online version of this article includes the following figure supplement(s) for figure 1:

**Figure supplement 1.** Split-finger protein 1 (Sfp1) binds Rpb4 and its efficient import is dependent on *RPB4*.

---

cell size and wild-type growth (*Figure 2—figure supplement 1A*, results not shown), indicating that the fusion protein is functional. The CRAC sequencing results were processed by a pipeline that Granneman's group has developed previously (*van Nues et al., 2017*). Metagene analysis of two biological replicates demonstrated a sharp peak toward the 3' end of the mRNAs, at a discrete relative distance (percentage wise) from polyadenylation sites (pAs) (*Figure 2A*). Since the two replicates were very similar, we combined them for further analyses.

To identify mRNA targets that Sfp1 binds to, we took the top 600 mRNAs with the highest number of CRAC reads and performed k-means clustering of the signal around the pAs (*Figure 2B*). We found four distinct clusters, three of which (C1-3) together accounted for 97.9% of the reads, distributed across 262 genes, spanning a region of 250 bp upstream of the pAs. The remaining mRNAs in the heatmap (C4), representing only 2.1% of the mapped CRAC reads, showed no clear accumulation of CRAC reads. The above 262 genes were named herein 'CRAC positive (CRAC+)' genes (*Supplementary file 2*). We next assigned a relative value, named CRAC index, to each CRAC + mRNA. This value reflects the number of CRAC RPKM normalized to its mRNA level (*Supplementary file 2*). Normalizing it to the ChIP-seq signal yielded similar results (see sheet 2 in *Supplementary file 2*). We assume that these values reflect the propensity of Sfp1 to co-transcriptionally bind Pol II transcripts. GO term enrichment analysis indicates that many CRAC + genes encode proteins related to protein biosynthesis (*Figure 2—figure supplement 1B*), including genes encoding translation factors, 70 ribosomal proteins (RP) ($p=7.7e^{-58}$), and 25 ribosome biogenesis (RiBi + RiBi like) proteins ($p=7.2e^{-11}$). 'RiBi-like' genes were previously defined as being activated by Sfp1, similarly to canonical RiBi. Specifically, Sfp1 binds to the upstream regions of both RiBi and Ribi-like genes and functions as an activator - as determined by anchor-away depletion followed by Pol II ChIP-seq (*Albert et al., 2019*). To obtain a more quantitative view, we crossed the list of CRAC + genes with a previously published list of genes whose expression is altered by either Sfp1 depletion or overexpression (*Albert et al., 2019*), and found that 42% of the CRAC + genes belong to a group whose transcription is upregulated by overexpressing Sfp1 ($p<2.2e^{-36}$) (*Figure 2—figure supplement 1C*). The significant overlap between mRNAs that are both transcriptionally regulated and physically bound by Sfp1 suggests a link between these two Sfp1 activities.

Next, we performed motif analysis using DRIMust and MEME tools (see M&M). We focused on C1-C2 clusters that show a clear peak upstream of the pA sites. Out of 113 mRNAs in these clusters, we found 73 mRNAs with a short motif (GCTGCT, some with more than two GCT repeats) ($p<7.925e^{-08}$), at an average distance of 150 bases upstream of the pA (*Figure 2—figure supplement 1F*). Metagene analysis showed that Sfp1 tends to bind mRNAs in close proximity to the motif (*Figure 2C*). The C3 cluster also contains many genes having similar motifs but not located at a distinct distance from the pA. Screen-shots of two representative genes with the GCTGCT motif are shown in *Figure 2—figure supplement 1G*.

To corroborate the interaction of Sfp1 with CRAC + mRNAs, we performed an RNA immunoprecipitation (RIP) analysis. Four out of the five mRNAs tested were co-immunoprecipitated with Sfp1. As a negative control, we selected *GPP1*, an mRNA not belonging to the 262 CRAC + mRNAs. Unlike the CRAC + mRNAs, *GPP1* mRNA was not immunoprecipitated by Sfp1 (*Figure 2D*, '*GPP1*'). As a positive control, we performed Rpb4-RIP and found that all five pre-mRNAs, including *GPP1*, were co-purified with Rpb4. Importantly, Sfp1 pulled down intron-containing *RPL30* pre-mRNA (*Figure 2D*). Note

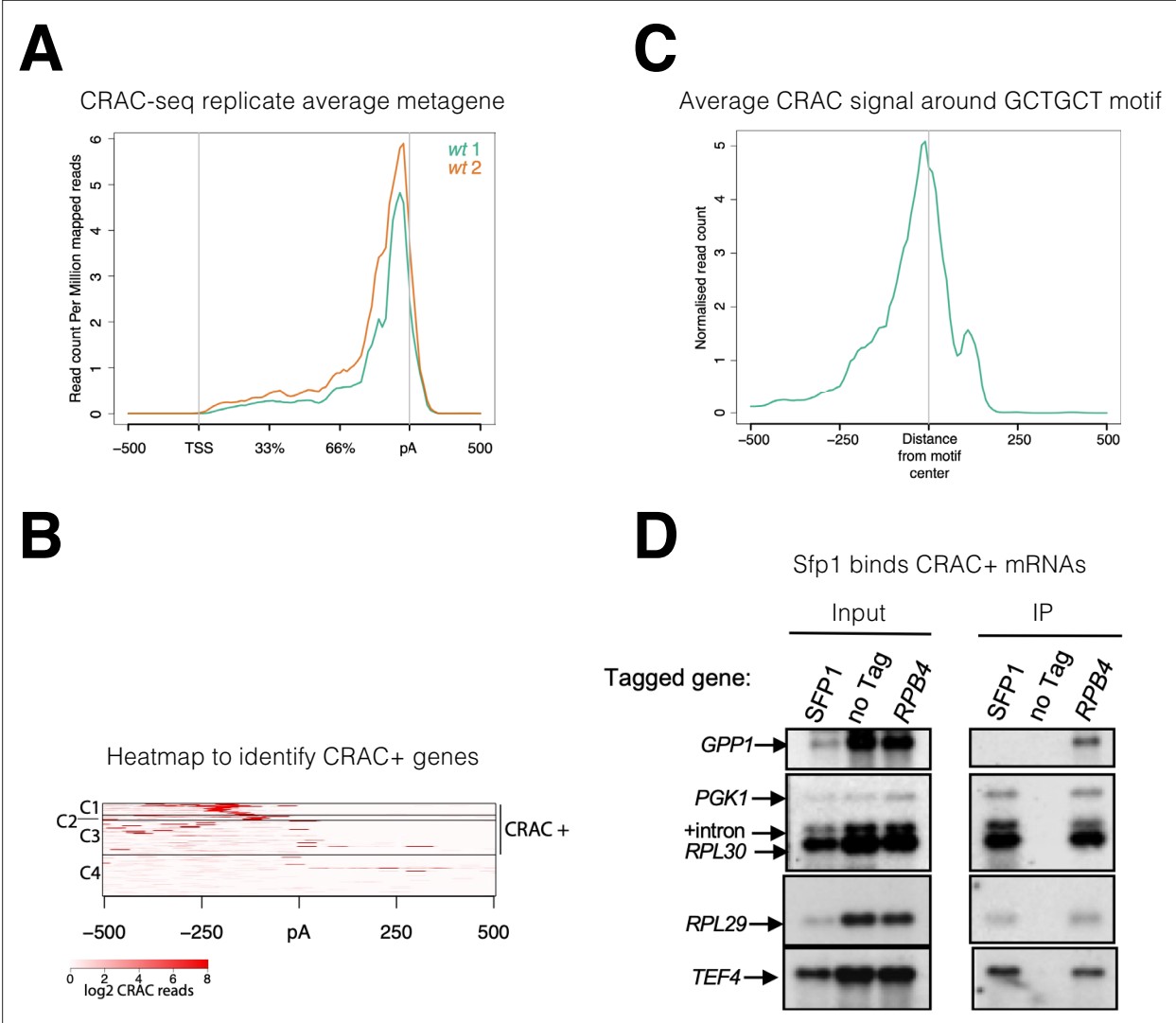

**Figure 2.** Split-finger protein 1 (Sfp1) binds a group of mRNAs around GCTGCT motif. (**A**) *Metagene profile of cross-linking and analysis of cDNA (CRAC) analysis in two wild-type replicates*. RPKMs plot around the average metagene region of all yeast genes. The 5'UTR and 3'UTR are shown in real scale in base pairs (bp) whereas the transcribed region is shown as a percentage scale to normalize different gene lengths. (**B**) *Heatmap representation of CRAC reads around the polyadenylation site (pA) for the top 600 genes with the highest number of CRAC reads*. The genes with high CRAC signal density upstream of the pA site are considered as CRAC+ (n=262), indicated on the right. The chosen cut-off was somewhat arbitrary; additional analyses shown in the subsequent figures indicate that this choice was biologically significant. (**C**) *Average metagene analysis of two replicates of the CRAC + signal in genes containing a GCTGCT motif in a region ±500 bp around the motif*. CRAC reads were aligned by the center of the motif. (**D**) *Sfp1 pulls down CRAC + mRNAs*. The extracts of isogenic cells, expressing the indicated tandem affinity purification tag (TAP), were subjected to tandem affinity purification (***Puig et al., 2001***), in the presence of RNase inhibitors. The RNA was extracted and was analyzed by Northern blot hybridization, using the probes indicated on the left. Note that the 'SFP1' lane in the input panel was underloaded. '+intron' denotes the position of intron-containing *RPL30* RNA.

The online version of this article includes the following figure supplement(s) for figure 2:

**Figure supplement 1.** Various features of *Sfp1-RNA* interaction.

that the ratio between pre-mRNA and mature mRNA is higher in the IPed lane than in the input one. Since introns are almost fully spliced out during transcription (*Wan et al., 2020*), these results strongly suggest that Sfp1 binds nascent RNA co-transcriptionally.

## Sfp1 RNA-binding capacity is regulated by promoter-located Rap1-binding sites

There is increasing evidence that promoters can affect mRNA stability, but the underlying mechanisms are unclear (*Dahan and Choder, 2013*; *Haimovich et al., 2013a*). Sfp1 is recruited to its cognate promoter by two alternative modes: either (i) directly through an A/T rich binding site or (ii) by Rap1 - directly or via Ifh1 (*Albert et al., 2019*; *Reja et al., 2015*). Previously, we reported that Rap1 also regulates the decay of its transcripts (*Bregman et al., 2011*), raising the possibility that this feature is mediated by Rap'1 capacity to recruit mRNA decay factor(s). To test the possibility that Rap1 function in mRNA decay is mediated by Sfp1 recruitment, we examined whether a Rap1 binding site (RapBS), coincided with Sfp1 DNA binding in many promoters (*Reja et al., 2015* and references therein), is required for Sfp1 RNA binding. To this end, we compared the binding of Sfp1 to mRNAs derived from two similar plasmids, which transcribe identical mRNAs (*Bregman et al., 2011*). Each construct contains the *RPL30* transcription unit, including the 3' noncoding region. *RPL30* mRNA is one of the CRAC + mRNAs that bind to Sfp1 (*Supplementary file 2* and *Figure 2D*). Transcription from both constructs is governed by the same TATA box (*Figure 3A*). One of our constructs ('construct A') contains an upstream activating sequence (UAS) that naturally lacks a RapBS (*ACT1*p), and the other ('construct B') contains a UAS that naturally contains 2 RapBSs (*RPL30*p). To differentiate plasmid-borne from endogenous *RPL30* transcripts, we introduced a tract of oligo(G)18 into the 3' noncoding sequence of the plasmid-derived *RPL30* genes. An RIP assay demonstrated that, despite being identical mRNAs (see *Bregman et al., 2011*), only the mRNA transcribed by construct B bound Sfp1 (*Figure 3B*). To determine whether RapBS is responsible for Sfp1 RNA binding, we first introduced two RapBS into construct A, creating 'construct E,' and found that the mRNA gained Sfp1-binding capacity (*Figure 3B*, construct E). We next deleted the two RapBS from their natural position in the *RPL30* promoter, creating 'construct F,' and found that the mRNA could no longer bind to Sfp1 (*Figure 3B*, construct F). Note that the RNA level of 'construct F' is higher than the level of 'construct E' (*Figure 3B*, 'Input' panel). Despite this difference, only the RNA encoded by construct E was detected in the IP panel. This clearly shows that the detection of the RNA was not merely a result of its expression level. Collectively, these results indicate that the capacity of Sfp1 to bind to RNA is mediated by promoter elements. RapBS is necessary and sufficient to mediate the interaction of Sfp1 with the gene transcript.

To obtain a more general perspective, we crossed the lists of (i) RiBi (+RiBi like) mRNAs that bind to Sfp1 (RiBi CRAC+) (25 genes), (ii) non-CRAC + RiBi (412 genes), and (iii) genes whose promoters bind Rap1, as demonstrated by ChIP-seq (*Lieb et al., 2001*; *Figure 3C*). While most RiBi non-CRAC+ promoters do not bind to Rap1, most RiBi CRAC+ do bind to Rap1 (p-value = $1.9 \times 10^{-22}$). Moreover, RiBi non-CRAC + exclude genes having Rap1 binding (*Figure 3C*; p-value of exclusion is $1.5 \times 10^{-5}$). In contrast, RP genes show no statistical relationship with Rap1 binding because all of them (except for 10 genes) contain RapBS (*Reja et al., 2015*). Thus, the correlation between the presence of RapBS in promoters and the capacity of Sfp1 to bind to mRNA is highly significant.

Sfp1 interacted also with endogenous RP mRNAs (*Figure 2D* and *Figure 3B*, lower panels). We also examined TAP-tagged Fhl1 and Ifh1 by RIP analysis. Unlike Sfp1, no interaction was detected between these two transcription factors and the examined mRNAs (*Figure 3B*, lower panels).

Taken together, these results are consistent with the possibility that recruitment of Sfp1 to its cognate promoters, mediated by Rap1, is necessary for its capacity to co-transcriptionally interact with the transcript.

## Sfp1 affects mRNA decay of CRAC+ genes

The interaction of Sfp1 with mRNA and with Rpb4, as well as its localization to P-bodies, suggests that Sfp1 plays a role in mRNA decay. To test the involvement of Sfp1 in mRNA synthesis and its possible role in mRNA decay, we performed a Genomic Run-On (GRO) assay (*García-Martínez et al., 2004*) in WT, *sfp1Δ,* and Sfp1-depleted cells. As expected, the deletion of *SFP1* affected the mRNA synthesis rates (SR) and abundances (RA) of the ribosomal biosynthetic (RiBi) and genes encoding ribosomal

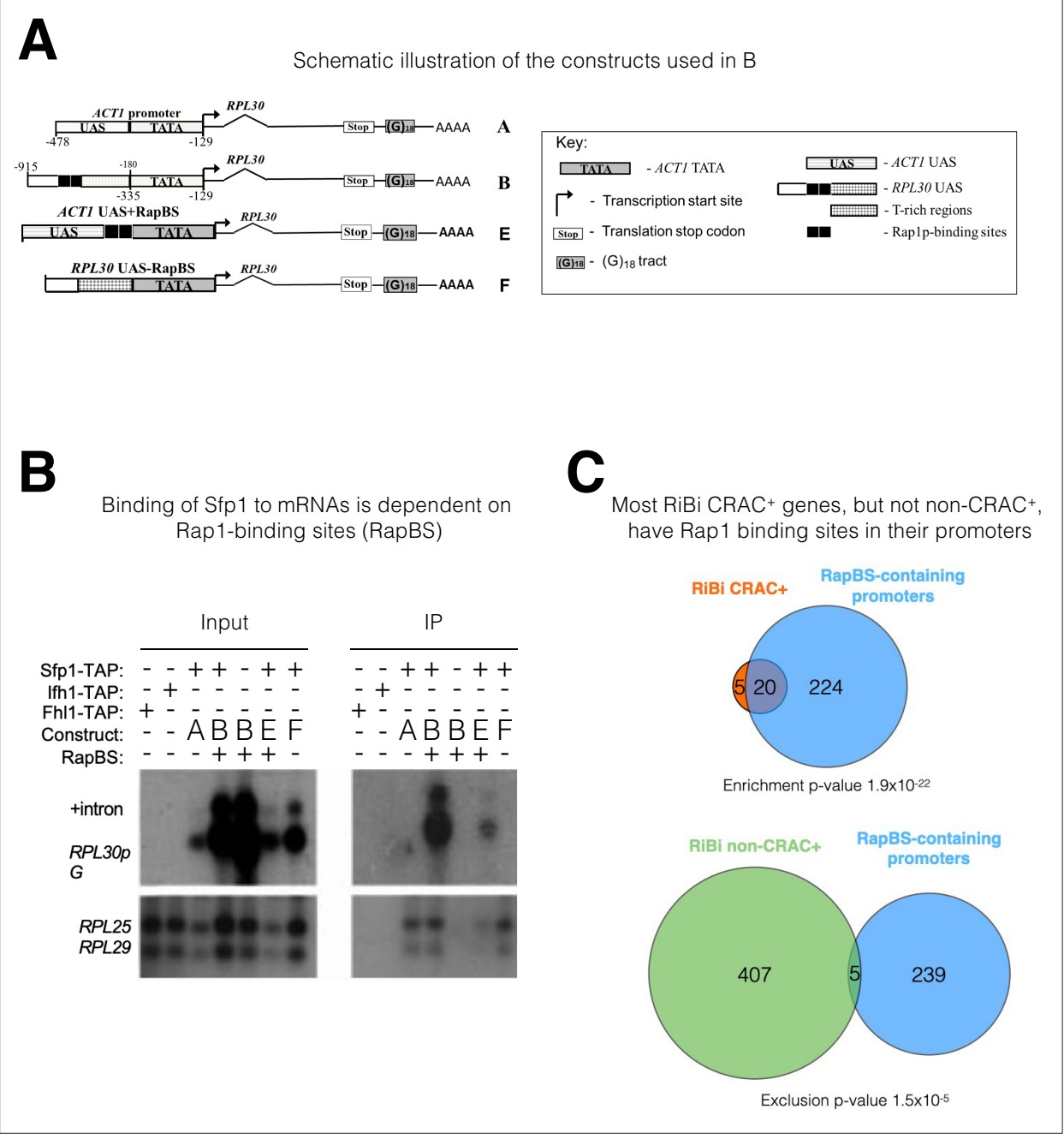

**Figure 3.** The mRNA-binding specificity of split-finger protein 1 (Sfp1) depends on the Rap1 binding site (RapBS) within the promoter. (**A**) *Constructs used in this study*. The constructs were described previously (***Bregman et al., 2011***). To differentiate construct-encoded mRNAs from endogenous ones, we inserted an oligo(G)$_{18}$ in the 3' untranslated region. The constructs are identical except for the nature of their upstream activating sequence (UAS), located upstream of the *ACT1* core promoter that includes the TATA box (designated 'TATA'). The nucleotide boundaries of the *RPL30* sequences are depicted above the constructs, and those of the *ACT1* sequences are depicted below the constructs. The numbering referes to the translation start codon. These constructs encode identical mRNA (***Bregman et al., 2011***). (**B**) *Binding of Sfp1 to mRNA is dependent on the RapBS*. Extracts of cells, expressing the indicated constructs, were subjected to RNA immunoprecipitation (RIP), using tandem affinity purification (TAP) of the indicated TAP-tagged proteins. RIP was followed by Northern blot hybridization using the probes indicated at the left. After the membrane was hybridized with oligo(C)$_{18}$-containing probes (to detect *RPL30pG* mRNA; see ***Bregman et al., 2011***), the membrane was hybridized with probes to detect endogenous *RPL25* and *RPL29* mRNAs.+Intron represents the intron-containing *RPL30pG* RNA (**C**) *Most RiBi CRAC + genes have Rap1 binding sites (RapBS) in their promoters*. Only 25 RiBi genes (including RiBi-like) are defined as CRAC+. Upper panel: Venn diagram showing the overlap between these genes and genes carrying promoters with RapBS. Lower panel: Venn diagram showing the overlap between all RiBi (+RiBi like) genes excluding CRAC+ ('non-CRAC+') and genes carrying promoters with RapBS. Data on Rap1 bound promoters were obtained from ***Lieb et al., 2001***. A hypergeometric test

*Figure 3 continued on next page*

Figure 3 continued

was applied to calculate the p-values, indicated underneath each diagram. Note that p-values were significant for both inclusion (upper diagram) and exclusion (lower diagram).

proteins (RP) (*Figure 4A*, see 'SR' and 'RA' panels). Calculating the mRNA half-lives (HL) revealed, unexpectedly, that deletion of Sfp1 led to reduced stability of CRAC + mRNAs (*Figure 4A*, 'HL' panel) including RP genes and RiBi CRAC+. Depletion of Sfp1 by the auxin-induced degron system (AID) (*Nishimura et al., 2009*) for 20 min resulted in a decrease in synthesis rate (SR) of CRAC + genes (*Figure 4B* left, see shift of green spots distribution to the left), consistent with previous results (*Albert et al., 2019*) and references therein. This resulted in a proportional decrease in mRNA abundance (RA) ratios (RA in auxin treated/RA before treatment) (Spearman coefficient = 0.59). Forty min later, we observed a larger decrease in SR and RA of many genes (*Figure 4B* right). RA of CRAC + mRNAs decreased more than expected from just an effect on transcription (*Figure 4B* right, the green spots are scattered below the correlation line), suggesting that they were degraded faster than the rest of the transcriptome. This conclusion was corroborated by examining degradation rates of single CRAC + mRNAs, either by comparing mRNA decay kinetics in Δsfp1 and in WT (*Figure 4C*, 'endogenous genes'), or at 1 hr after auxin addition (*Figure 4—figure supplement 1A*). As a control, we examined the degradation kinetics of mRNA with the lowest CRAC index (PMA1) and observed little effect of Sfp1. To further corroborate the high-throughput results, we took an additional single-gene approach. In this approach, we measured HL by naturally blocking transcription after shifting cells, which had proliferated at 30°C, to 42 °C (*Lotan et al., 2005* and references therein), and determined the rate of decrease in mRNA as a function of time post transcription arrest. We monitored the HL of two mRNAs with high CRAC index, *RPL28* and *RPL29* mRNAs. We chose *PMA1* and *GPP1* to represent mRNAs with low or no CRAC index, respectively. Consistently with the other methods we used to determine HL, also in this case, we observed a substantial shortening of *RPL28* and *RPL29* mRNA HLs due to *SFP1* deletion (*Figure 4—figure supplement 1B*, *RPL28*, and *RPL29*). Previously, the levels of RP mRNAs were reported to decrease rapidly in response to HS (*Bresson et al., 2020*). Here, we corroborated these results and showed that, in the absence of *SFP1*, this decrease occurs more rapidly. In contrast, the HLs of *PMA1* and *GPP1* mRNAs were not affected by *SFP1* deletion (*Figure 4—figure supplement 1A*, *PMA1*, and *GPP1*).

Consistent with Sfp1-binding as a prerequisite feature for its capacity to stabilize mRNAs, the calculated HLs of RP mRNAs and the 25 RiBi CRAC+, but not RiBi non-CRAC + mRNAs and 'Rest' (the bulk of the mRNAs), decreased more than those of the other genes due to *SFP1* deletion (*Figure 4A*, 'HL' panel). Remarkably, although Sfp1 binds all RiBi promoters and stimulates their transcription (*Albert et al., 2019*), it only affects the stability of a small subset of them – the CRAC + ones (see *Figure 4A* HL panel 'RiBi CRAC+'). Taken together, our data indicate that Sfp1 specifically stabilizes CRAC + mRNAs.

Since RapBS is required for the binding of Sfp1 to some mRNAs (*Figure 3B*), we examined whether RapBS was also required for Sfp1-mediated mRNA stability. To this end, we analyzed the effect of deleting *SFP1* on the stability of these identical plasmid-borne mRNAs. Sfp1 did not affect the stability of an mRNA whose synthesis was driven by a promoter that lacks RapBS (*Figure 4C*, left panel); yet, introducing just RapBS to the same promoter resulted in a transcript whose stability was dependent on Sfp1 (*Figure 4C*, right panel).

The first step in mRNA decay is the shortening of the mRNA poly(A) tail (*Parker, 2012*). Polyacrylamide gel electrophoresis northern (PAGEN) has been used to determine deadenylation rates (*Lotan et al., 2005*; *Sachs and Davis, 1989*). Using this technique, we found that Sfp1 slows down the mRNA deadenylation step of a few CRAC + mRNAs, as well as their subsequent degradation (*Figure 4D*). These findings suggest that Sfp1 plays a role in deadenylation and decay of CRAC + mRNAs.

Collectively, we found that the binding of Sfp1 to mRNAs slows down their deadenylation and stabilizes them. Promoter binding of Sfp1 via Rap1 is necessary for its mRNA binding (*Figure 3*) and mRNA stability (*Figure 4C*); however, since Sfp1 binds most RiBi promoters, not via Rap1 (*Figure 3C*), but not RiBi transcripts (most RiBi are not CRAC+, see *Figure 3C*) and does not affect HLs of most RiBi mRNA (*Figure 4A*), promoter binding is not sufficient for its mRNA binding and mRNA degradation activities. Perhaps, promoter binding of Sfp1 via Rap1 is necessary and sufficient for its RNA binding.

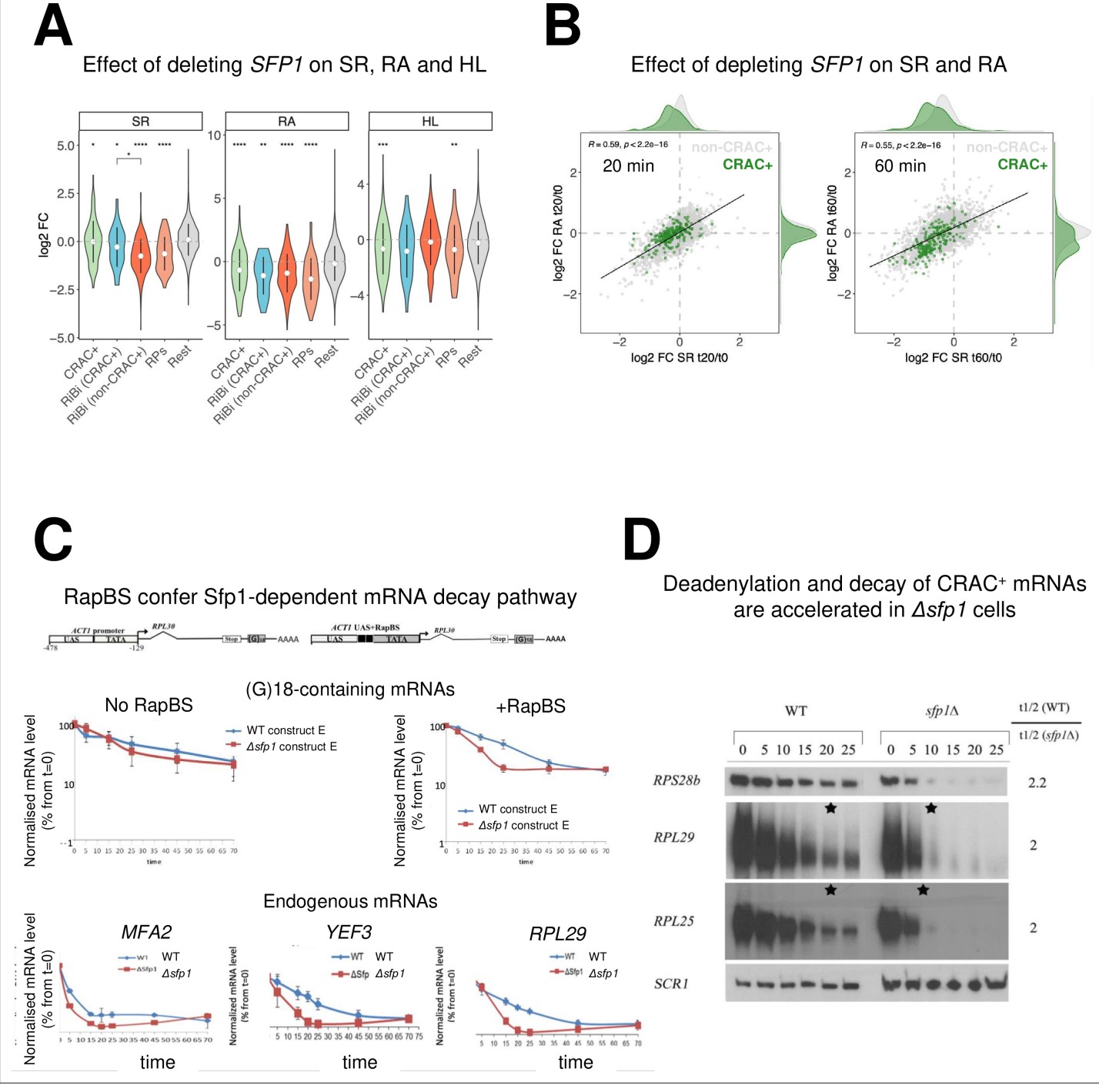

**Figure 4.** Split-finger protein 1 (Sfp1) is required for efficient transcription of CRAC + genes and stabilizes the deadenylation-dependent pathway of their mRNAs' decay. (**A**) *Sfp1 deletion affects SR (RNA synthesis rate), RA (mRNA Abundance), and HL (mRNA half-life) differently for different subsets of genes*. CRAC+ (n=262), RiBi (including RiBi-like) CRAC+ (n=25), RiBi (+RiBi like) excluding CRAC+ (RiBi (non-CRAC+)) (n=411), RPs (n=129)). Statistical analyses were performed using the Wilcoxon test. Asterisks indicate significant results (*p<0.05; **p<0.01; ***p<0.001; ****p<0.0001). Unless indicated otherwise, statistical comparisons were performed using the 'Rest' group (all detectable genes excluding CRAC + ones) as a reference. In addition, RiBi CRAC + and RiBi non-CRAC +are compared against each other. (**B**) *Sfp1 depletion rapidly affects SR and RA of CRAC + genes*. Scatterplot of changes in SR vs changes in RA at 20 min. (left) or 60 min. (right) after depleting Sfp1-degron by auxin. CRAC + genes are highlighted in green. Spearman correlation values and the significance of the linear adjustment for the whole dataset are indicated inside the plot. Density curves are drawn on the margins of the plot to help evaluate the overlap between dots. (**C**) *RapBS confers the Sfp1-dependent mRNA decay pathway*. Shown is the quantification of Northern blot hybridization results of mRNA decay assay (Methods), performed with wild-type (WT) or Δ*sfp1* cells that carried

*Figure 4 continued on next page*

Figure 4 continued

the indicated constructs (described in **Bregman et al., 2011** and schematically shown at the top). The membrane was probed sequentially with an oligo(C)$_{18}$-containing probe, to detect the construct-encoded mRNA, and with probes to detect endogenous mRNAs. mRNA levels were normalized to the Pol III transcript *SCR1* mRNA (Methods). The band intensity at time 0, before transcription inhibition, was defined as 100% and the intensities at the other time points (min) were calculated relative to time 0. Error bars indicate the standard deviation of the mean values of three independent replicates (for (**G**)$_{18}$-containing mRNAs), or of 12 replicates (for endogenous mRNAs). (**D**) The *deadenylation rate of CRAC + mRNAs and the subsequent decay of the deadenylated RNAs is accelerated in sfp1Δ cells*. Transcription was blocked as described in section C (Methods). RNA samples were analyzed using the polyacrylamide Northern technique (**Sachs and Davis, 1989**), using the probes indicated on the left, all are CRAC+. Half-lives were determined and the indicated ratios are depicted on the right. The asterisk (*) indicates the time point at which deadenylation is estimated to be complete.

The online version of this article includes the following figure supplement(s) for figure 4:

**Figure supplement 1.** Split-finger protein 1 (Sfp1) stabilizes CRAC + mRNAs.

**Figure supplement 2.** Split-finger protein 1 (Sfp1) binds CRAC + gene bodies.

## Sfp1-bound mRNAs are transcribed by Sfp1-bound genes in a manner that suggests a mechanistic linkage

Sfp1 is a well-known transcription factor that binds to promoters of its cognate genes (see Introduction). Our discovery that Sfp1 also binds to CRAC + mRNAs, which are only encoded by a fraction of the Sfp1 target genes (see previous sections), prompted us to investigate whether the chromatin binding feature of Sfp1 with CRAC + genes is different from its binding with other gene targets. To do so, we leveraged published ChIP-exo datasets (**Reja et al., 2015**) and found that Sfp1 binds not only to promoters but also to gene bodies of CRAC + genes (**Figure 5A**). In contrast, Rap1 binds almost exclusively to promoters of CRAC + genes (**Figure 5A** and **Figure 4—figure supplement 2A**). ChIP-exo data of a random subset of non-CRAC + genes showed that Sfp1 binds weakly to both promoters and gene bodies (**Figure 5A**, 'Sfp1 control'). However, because promoter binding is not higher than binding to other chromatin regions, it is unclear whether this binding is specific (but see below).

Importantly, the intensity of Sfp1 binding to the bodies of CRAC + genes correlates with the number of transcriptionally active Pol II per gene (this is what BioGRO-seq measures) (**Figure 5B**, left), suggesting that the binding of Sfp1 to gene bodies occurs through the binding of active Pol II. However, in addition to active Pol II, the chromatin accommodates transcriptionally arrested Pol II, most of which is presumed to be in a backtracked configuration (see Introduction). If, in addition to its binding to active Pol II, Sfp1 remains bound to backtracked Pol II, it is expected that the correlation of Sfp1 occupancy with the occupancy of all Pol II molecules would be even higher than it is with just active Pol II. Comparing the left and right panels of **Figure 5B** indicates that this is the case. Interestingly, we found a modest correlation between Sfp1 binding to the bodies of all genes (excluding CRAC+) and their BioGRO-seq signal (**Figure 5B**, left panel). This weak binding could indicate non-specific attachment to accessible chromatin, which characterizes transcriptionally active regions. However, because over-expression of Sfp1 affects the transcription of ~30% of the genes (**Albert et al., 2019**), we suspect that the weak binding of Sfp1 to the bodies of these genes is also through Pol II interaction.

**Albert et al., 2019** discovered that the complete set of ~500 gene promoters bound by Sfp1 could be revealed only by a combination of ChIP-seq and ChEC-seq ('chromatin endogenous cleavage,' using micrococcal nuclease [MNase] fused to Sfp1) methods. These two methods revealed the two distinct promoter binding modes of Sfp1, discussed earlier. We compared the Sfp1 ChIP-exo and ChEC-seq metagene profiles, which were obtained in different laboratories, and found, in both methods, substantial differences between Sfp1 binding to CRAC + and non-CRAC + genes (**Figure 4—figure supplement 2B**). Specifically, in both methods, there is a substantial difference between CRAC + and non-CRAC+. In ChIP-exo the difference is considerable. Focusing on ChEC-seq profiles, there seems to be a difference in Sfp1 positions in the CRAC + promoters relative to the TSS with that of non-CRAC+ (**Figure 4—figure supplement 2B**, see the centers of the orange and the red peaks). Taken together, these results suggest that the binding of Sfp1 to CRAC + promoters is different from that to non-CRAC + promoters. This difference is in addition to that found in gene bodies, observed by ChIP-exo (**Figure 5A**, **Figure 4—figure supplement 2B**).

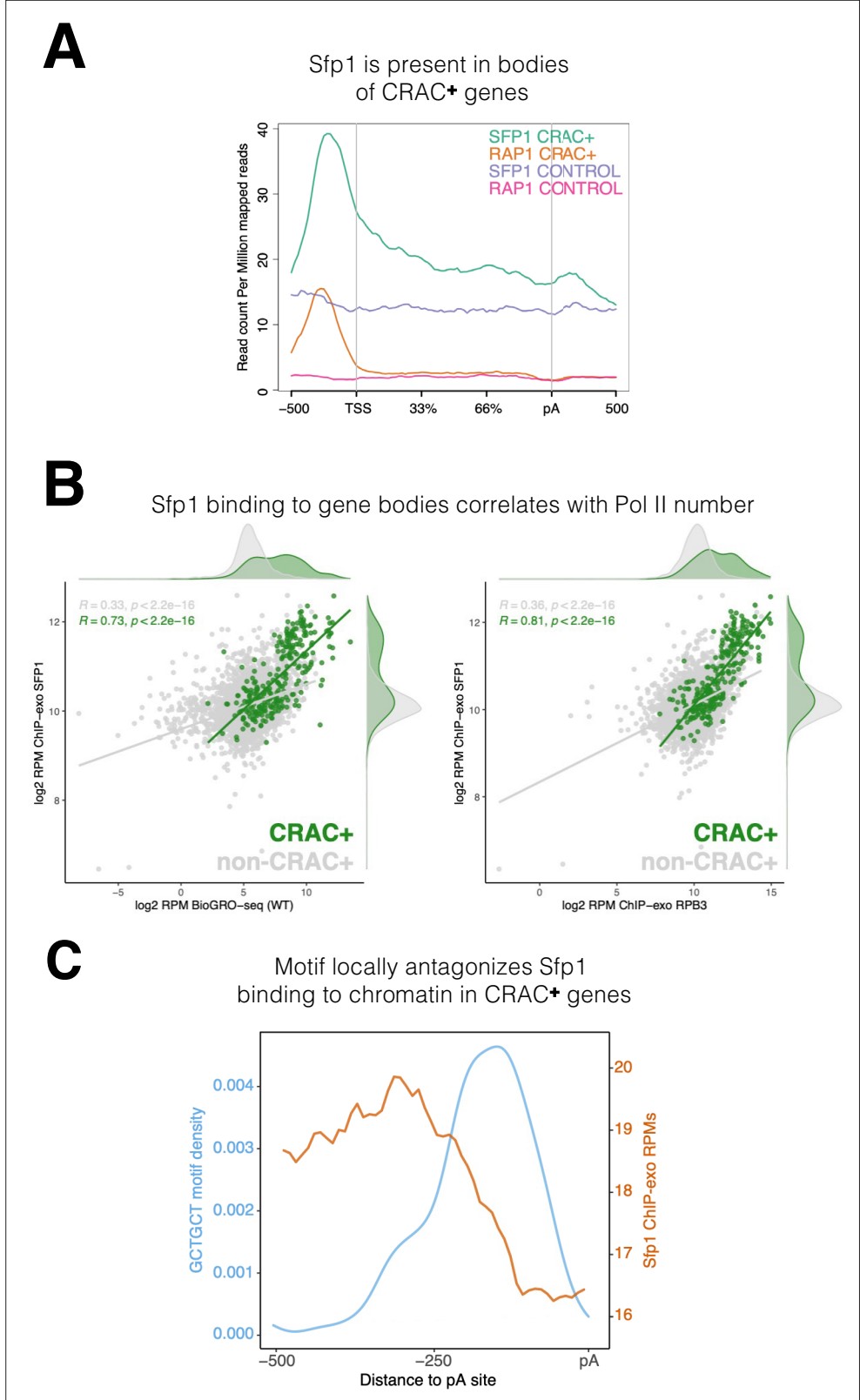

**Figure 5.** Binding features of split-finger protein 1 (Sfp1) to chromatin. (**A**) *Sfp1 is present in the bodies of CRAC + genes*. Average metagene of Sfp1 and Rap1 chromatin immunoprecipitation (ChIP)-exo signal, obtained from *Reja et al., 2015*, for CRAC + genes (n=262) and CRAC-CONTROL genes (a subset of 262 non-CRAC + genes randomly selected from the entire genome, but excluding ribosomal protein (RP) and ribosome Biogenesis

*Figure 5 continued on next page*

*Figure 5 continued*

biogenesis (RiBi) genes). See also the plot with alternative scaling in *Figure 4—figure supplement 2A*. (**B**) Left panel: *Positive correlation between Sfp1 binding to gene bodies and transcriptional activity*. Scatterplot comparing Sfp1 binding to gene bodies (measured by ChIP-exo, *Reja et al., 2015*) versus the density of actively elongating RNA polymerase II (pol II) (measured by BioGRO-seq; *Begley et al., 2021*) in CRAC + genes (n=262). Spearman correlation is indicated at the bottom of the plot. Right panel: *Correlation between Sfp1 binding to the bodies of all genes (n=4777 that were in the dataset) and transcription rate*. Spearman correlation for all genes (including CRAC+) or for all genes excluding CRAC + genes are indicated at the top of the plot. (**C**) The *Sfp1 ChIP-exo signal drops downstream of the GCTGCT motif*. Comparison of average metagene profiles of cross-linking and analysis of cDNA (CRAC) (blue) and ChIP-exo (orange) signals for genes with a GCTGCT motif (n=163).

The presence of Sfp1 along gene bodies of CRAC + genes raised a possibility that this feature could be related to its capacity to bind their transcripts. To explore this possibility, we examined the Sfp1 ChIP-exo signal of CRAC + genes from –500 bp up to the pA sites. We found a decline in the signal that coincided with the position of the GCTGCT (*Figure 5C*), which is mapped ~160 bp upstream of the pA sites (*Figure 2—figure supplement 1F*). To obtain a wider view, we examined the ChIP-exo signal at ± 500 bp around the GCTGCT motif and observed a similar drop in the signal around the motif (*Figure 2—figure supplement 1D*). We also examined non-CRAC + genes and found a decline of the Sfp1 ChIP-exo signal around the the pAs. However, this decline was milder than that of CRAC + and occurs closer to the pAs (*Figure 2—figure supplement 1E*). These results, in combination with the dependence of Sfp1-mRNA binding on RapBS (*Figure 3B–C*), strongly suggest that Sfp1 dissociates from the chromatin and binds the RNA co-transcriptionally. Other observations that are consistent with co-transcriptional binding are outlined in the Discussion.

## Genes transcribing Sfp1-bound ('CRAC+') mRNAs exhibit higher levels of Pol II backtracking

Sfp1 negatively affects transcription elongation, as deletion of *SFP1* results in a higher Pol II elongation rate (*Begley et al., 2019*). Consistently, Sfp1 seems to enhance Pol II backtracking as a deletion of *SFP1* suppresses the effect of TFIIS deletion on transcription (*Gómez-Herreros et al., 2012a*). A characteristic feature of backtracked Pol II is the displacement of nascent RNA 3′ end from the active site (*Cheung and Cramer, 2011*). Consequently, backtracked Pol II cannot elongate transcription in GRO assays (*Jordán-Pla et al., 2015*; *Pelechano et al., 2009*). Therefore, a Pol II backtracking index (BI) can be determined by comparing Pol II ChIP (total Pol II) and GRO (actively elongating Pol II – not backtracked) signals. To determine the impact of Sfp1 on backtracking, we performed Rpb3 ChIP and GRO analyses in WT and *sfp1Δ* cells. As expected from a TF that stimulates transcription initiation, Rpb3 ChIP signals were higher in WT than in *sfp1Δ* cells (y=0.578 x) (*Figure 6A*, left). However, despite different Pol II occupancy, GRO signals were similar in WT and *sfp1Δ* strains (y=0.844 x) (*Figure 6A*, right), consistent with Sfp1-mediated stimulation of Pol II backtracking (in WT cells). This effect was particularly intense in genes whose mRNAs bind Sfp1 (CRAC+) (green dots in *Figure 6A*). Consequently, BI values of CRAC +genes were higher than the average (*Figure 6B*), suggesting a linkage between backtracking and Sfp1 imprinting (see below). Perhaps, the backtracked configuration is compatible with the movement of Sfp1 from Pol II to the nascent transcript (see Discussion).

We note that CRAC + genes cover a wide range of transcription levels (see *Figure 6A*), indicating that BI and transcription levels are unrelated. Moreover, among CRAC + genes, BI of RiBi genes whose transcripts bind Sfp1 (RiBi CRAC+) exhibited high BI with strong dependence on Sfp1, whereas RiBi genes that encode mRNAs that are not bound by Sfp1 ('RiBi non-CRAC+') did not (*Figure 6B*).

Sfp1 disruption affects both mRNA HL (*Figures 4 and 6C*) and BI (*Figure 6B and C*). In both cases, these effects were significantly stronger for the CRAC + group. We were intrigued by a possible link between these two effects, as it raises a possible role for backtracking in imprinting-mediated mRNA decay. In favor of this idea, we found a modest correlation between the effect of Sfp1 on HLs and on BIs (*Figure 6D*; r=0.366), which was higher for the CRAC + genes (*Figure 6D*: green spots, r=0.423). This indicates that the interaction of Sfp1 with chromatin has an effect on both Pol II elongation and mRNA HLs, suggesting a mechanistic linkage.

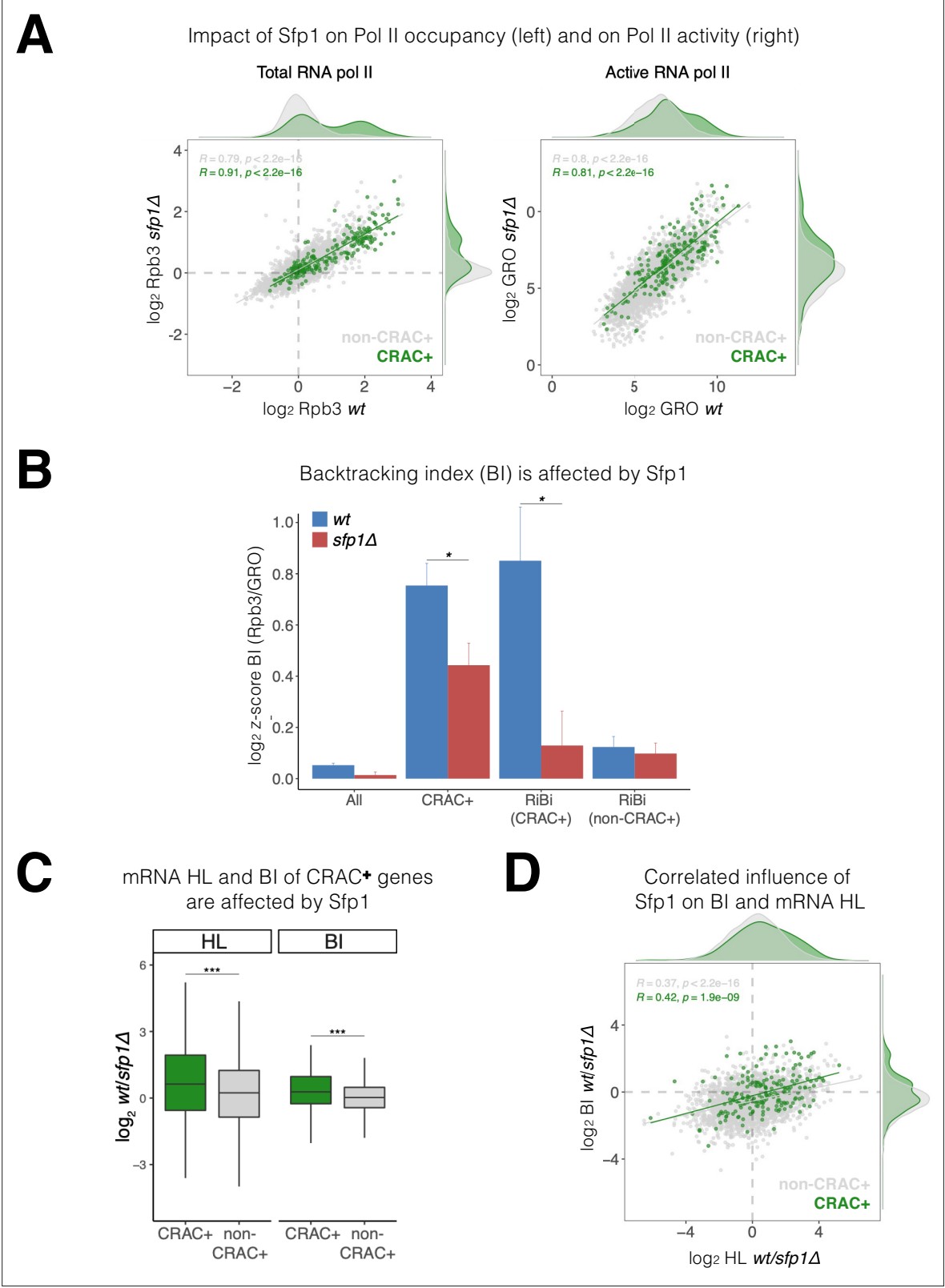

**Figure 6.** Split-finger protein 1 (Sfp1) induces polymerase II (Pol II) backtracking, preferentially in CRAC + genes. (**A**) *Sfp1 differently affects Pol II occupancy (left) and Pol II activity (right)*. Pol II levels were measured by Rpb3 chromatin immunoprecipitation (ChIP) and its activity by genomic run-on (GRO), in wild-type (WT) (BY4741) and its isogenic *sfp1Δ* strain, growing exponentially in YPD. Anti Rpb3 (rabbit polyclonal) ChIP on chip experiments using Affymetrix GeneChip *S. cerevisiae* Tiling 1.0 R custom arrays as described in Methods. For each gene, the average of signals corresponding to

*Figure 6 continued on next page*

Figure 6 continued

tiles covering 5' and 3' ends (250 bp) were calculated. Green dots represent the CRAC + genes; gray dot – non-CRAC+. The tendency line, its equation, Pearson R, and its p-value of the statistically significant deviation from the null hypothesis of no correlation are shown in gray, for the whole dataset, and in green, for the CRAC + genes. (**B**) *Sfp1 promotes Pol II backtracking of CRAC +genes*. Backtracking index (BI), defined as the ratio of Rpb3-ChIP to GRO signals, is shown for different gene sets, indicated below, comparing WT and *sfp1Δ* strains. In order to compare data obtained from different types of experiments the values were normalized by the median and standard deviation (z-score). The bars represent standard errors. Statistical significance of the differences between the averages of the indicated samples was estimated using a two-tailed Student's t-test (* means <0.01). (**C**) *mRNA HL and BI of CRAC + mRNAs/genes are affected by Sfp1*. Box and whisker plots showing the effect of Sfp1 on mRNA HL and Pol II BI. A comparison between CRAC+ (green) and all genes excluding CRAC + ones ('non-CRAC+' gray) genes is shown. Half-lives (HL) were calculated from the mRNA abundance (RA) and synthesis rates (SR), using the data shown in *Figure 4A*. The statistical significance of the differences between the averages of the CRAC + and non-CRAC + was estimated using a two-tailed Student's t-test (***p<0.0001). (**D**) *mRNA HL and BI are correlated via Sfp1: correlation between Sfp1-dependence of BI and mRNA HL ratios*. Data from C were represented in a scatter plot. Linear regression equations are shown for all (gray) and CRAC + genes (green). Pearson correlation coefficient, *R*, and the p-value of the statistically significant deviation from the null hypothesis of no correlation (*R=0*) are also indicated. All statistical correlations were determined using the ggpubr package in R.

## Sfp1 locally changes Rpb4 stoichiometry or configuration during transcription elongation

The yeast Rpb4 is a sub-stoichiometric Pol II subunit (*Choder, 2004*; *Choder and Young, 1993*), which enhances Pol II polymerization activity (*Fischer et al., 2020*; *Rosenheck and Choder, 1998*) and promotes mRNA instability due to co-transcriptional imprinting (*Goler-Baron et al., 2008*; *Lotan et al., 2007*; *Lotan et al., 2005*). Since Sfp1, which physically interacts with Rpb4 (*Figure 1—figure supplement 1A*), also affects mRNAs stability via imprinting (*Figures 2D and 4*), we examined whether Sfp1 and Rpb4 activities are interrelated. As a proxy for Rpb4 stoichiometry, we calculated the Rpb4-ChIP/Rpb3-ChIP ratio and observed a gradual decrease as Pol II traversed past the TSS, decreasing even further towards transcript pAs (*Figure 7A*, WT). These data suggest that Rpb4 gradually dissociates from Pol II during elongation (possibly concomitantly with transcript binding). However, this result could also be attributed to changes during transcription elongation in the capability of Rpb4 to contact DNA, or to changes in ChIP antibody accessibility that occur during elongation (referred to herein as 'Rpb4 configuration'). Global changes in Rpb4-ChIP/Rpb3-ChIP ratio were stronger in the highly transcribed genes in WT (*Figure 7A* cf 'Q1' and 'Q4'), suggesting a linkage to Pol II activity; i.e., the more active Pol II is the more likely it is to gradually lose Rpb4. Interestingly, the CRAC + genes exhibited an even stronger change in Rpb4 stoichiometry/configuration (*Figure 7A*, CRAC+) despite not being highly transcribed in most cases (*Figure 6A*), suggesting that this Rpb4 feature is modulated by Sfp1 molecules that are capable of binding CRAC + mRNA. Indeed, these changes in Rpb4 stoichiometry/configuration were highly dependent on Sfp1, as they were abolished in *sfp1Δ* for all gene sets analyzed (*Figure 7A*, *sfp1Δ*). Again, the effect of *sfp1Δ* was more pronounced in CRAC + than in non-CRAC + genes (*Figure 7A*, all panels). In summary, we detected a general Sfp1-dependent alteration in Rpb4 stoichiometry/configuration with Pol II during elongation, which was more pronounced in CRAC + genes.

We also found a general inverse correlation between the Rpb4-ChIP/Rpb3-ChIP ratio and BI in the WT, which was stronger in CRAC + genes (*Figure 7B*), suggesting that Rpb4 stoichiometry/configuration might affect Pol II backtracking, or vice versa–backtracking enhances changes in Rpb4 stoichiometry/configuration. This correlation was clearly reduced in *sfp1Δ* cells, which also showed almost no difference between CRAC + and non-CRAC +genes (*Figure 7B* cf left and right panels). Accordingly, the correlation between Sfp1 effects on BI and Rpb4-ChIP/Rpb3-ChIP ratio was tripled in CRAC + than in non-CRAC + genes (*Figure 7—figure supplement 1A*). These results suggest that Sfp1 activity during transcription elongation, Pol II backtracking, and Rpb4 stoichiometry/configuration are connected.

Finally, focusing on the CRAC + genes in the WT strain, we found a modest correlation between Rpb4 ChIP/Rpb3 ChIP ratio and mRNA HL (*Figure 7C*, *r*=0.488). Interestingly, this correlation was specific to CRAC + genes (very little correlation was found for the rest of the genes marked by gray spots) and was dependent on Sfp1 (*Figure 7C*, *sfp1Δ*). Accordingly, the correlation between Sfp1 effects on mRNA HL and Rpb4-ChIP/Rpb3-ChIP ratio was significant in CRAC +but absent in non-CRAC + genes (*Figure 7—figure supplement 1B*). Therefore, the link between Rpb4 stoichiometry/configuration and mRNAs HLs of CRAC + genes is entirely dependent on Sfp1, whereas the

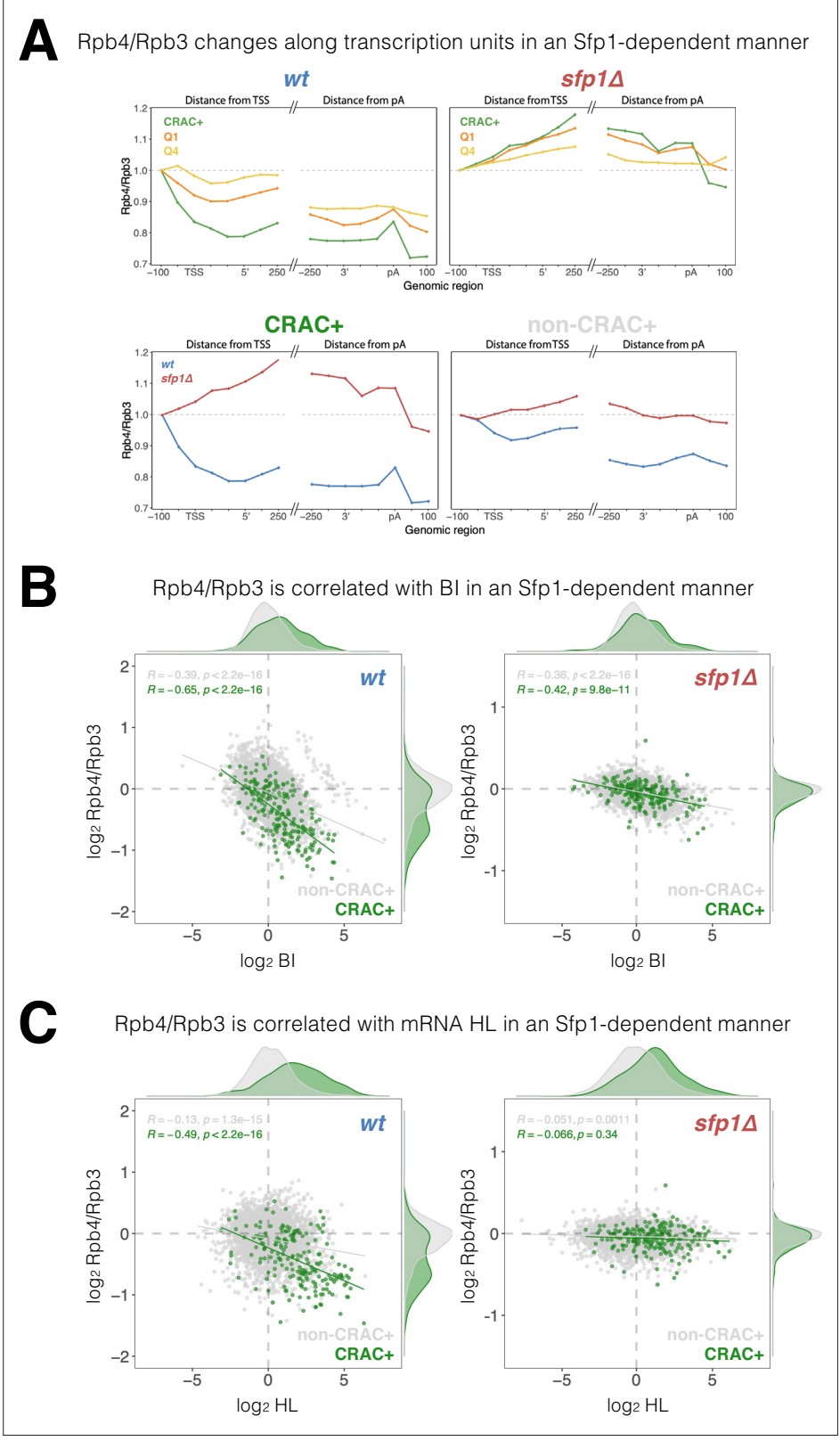

**Figure 7.** Split-finger protein 1 (Sfp) alters Rpb4 stoichiometry/configuration within the polymerase II (Pol II) elongation complex and this alteration is linked to mRNA stabilization. (**A**) *The Rpb4 stoichiometry/configuration changes along the transcription units in an Sfp1-mediated manner*. Top left panel - the values of Rpb3 and Rpb4 were obtained from chromatin immunoprecipitation (ChIP) on chip experiments, either against Rpb3 or against

*Figure 7 continued on next page*

*Figure 7 continued*

Rpb4-Myc in LMY3.1 cells proliferated exponentially in YPD. Rpb4-ChIP/ Rpb3-ChIP ratios were calculated and averages for the indicated gene sets were obtained for positions ranging from –100 to +250 (relative to TSS) and from –250 to +100 (relative to pA sites). Average ratios were normalised to the TSS –100 position, in order to represent profiles of Rpb4-ChIP changes after Pol II recruitment to promoters. Top right panel –profiles of Rpb4-ChIP/Rpb3-ChIP ratios were obtained as in the top left panel, but from *sfp1Δ* strain (LMY7.1). Bottom left panel - Rpb4-ChIP/Rpb3-ChIP profiles of CRAC + genes: comparing WT (blue) and *sfp1Δ* (red) strains. Bottom right panel - Rpb4-ChIP/Rpb3-ChIP profiles of rest of the genes (all detectable genes excluding CRAC + ones): comparing WT (blue) and *sfp1Δ* (red) strains. (**B**) *Correlation between the Rpb4-ChIP/Rpb3-ChIP ratios and Pol II BI in WT (left panel) or in sfp1Δ cells (right panel).* For each gene, the average of Rpb4-ChIP/Rpb3-ChIP values corresponding to positions from TSS to + 250 and from –250 to pA sites were calculated. BI values are taken from **Figure 6C**. Linear regression equations are shown for all (gray) and CRAC + genes (green). Pearson correlation coefficients, *R*, and the p-values of the statistically significant deviation from the null hypothesis of no correlation (*R*=0), are also shown. All statistical correlations were determined using the *ggpubr* package in R. (**C**) *Correlation between the Rpb4-ChIP/Rpb3-ChIP ratios and mRNA HL in WT or sfp1Δ cells.* Values of Rpb4-ChIP/Rpb3-ChIP ratios ere determined as in B. mRNA HL was indirectly calculated from mRNA abundance and transcription rates taken from the data used in **Figure 4A** and is shown in arbitrary units. CRAC + genes are depicted in green. R was calculated as in B.

The online version of this article includes the following figure supplement(s) for figure 7:

**Figure supplement 1.** Correlation between the effects of split-finger protein 1 (Sfp1) on Rpb4-ChIP/Rpb3-ChIP ratios, on RNA polymerase II (Pol II) BI, and on mRNA HL.

---

correlation between the Rpb4-ChIP/Rpb3-ChIP ratio and BI is still present and significant in *sfp1Δ* (**Figure 7B**). This suggests that the impact of Sfp1 on mRNA stability is preferentially mediated by Rpb4 association, rather than by Pol II backtracking.

Taken together, we propose that the effects of Sfp1 on mRNA stability and on Pol II elongation are linked. Our results are compatible with a model whereby Sfp1 impact on Rpb4 during elongation provokes mRNA imprinting, which affects stability, and concomitantly produces Pol II backtracking.

## Discussion

Sfp1 has traditionally been regarded as a 'classical' TF known to bind specific promoters, either through interactions with other TFs or by directly binding to promoter DNA (**Albert et al., 2019**; **Reja et al., 2015**). Further analysis that we performed using published ChIP-exo and ChEC-seq data has revealed an additional aspect of Sfp1's binding behavior—it also interacts with gene bodies. Notably, the binding of Sfp1 to CRAC + gene bodies correlates with the presence of transcriptionally active Pol II molecules (**Figure 5B**), suggesting a physical association between Sfp1 and Pol II. Consequently, the binding of Sfp1 has an impact on Pol II configuration, stoichiometry, and backtracking, particularly in the context of CRAC + genes (**Figures 6 and 7**). This alteration in the Pol II configuration is reflected in the increased backtracking frequency, as indicated by the BI values. Moreover, the binding of Sfp1 to gene bodies correlates with Pol II activity, particularly within CRAC + genes (**Figure 5B**). Thus, Sfp1's function in a subset of its target genes extends beyond transcription initiation and encompasses transcription elongation, as we proposed earlier (**Gómez-Herreros et al., 2012b**). Based on the following observations, we propose that Sfp1 binds to Pol II in proximity to both DNA and Rpb4, accompanying Pol II during elongation and influencing its configuration, thereby enhancing its propensity to backtrack (**Figure 8A**): (I) Sfp1 physically interacts with Rpb4, either directly or indirectly (**Figure 1—figure supplement 1**). (II) ChIP-exo results, dependent on Sfp1's cross-linking with DNA, indicate Sfp1's presence near DNA, both at promoters and gene bodies (**Figure 5A**). (III) Sfp1 influences Pol II configuration in a manner that impacts the architecture or stoichiometry of Rpb4 within Pol II (**Figure 7**). (IV) Sfp1 binding to chromatin better correlates with all Pol II (both backtracked and active) than with transcriptionally active Pol II, examined by BioGRO-seq (**Figure 5B**). (V) The global reduction in BI is observed upon the deletion of *SFP1* (**Figures 6 and 7**).

The effects of Sfp1 on backtracking occur in the context of Rpb4 (**Figure 7B**). We note that backtracking is influenced by the alteration of Rpb4 within the elongation complex, independently of Sfp1, as the correlation between Rpb4 ChIP/Rpb3 ChIP ratios and BI is still present, albeit more mildly, in *sfp1Δ*. Rpb4/7 was previously implicated in backtracking either by itself (**Fischer et al., 2020**) or in the context of the Ccr4-NOT complex (**Babbarwal et al., 2014**; **Kruk et al., 2011**). It is possible

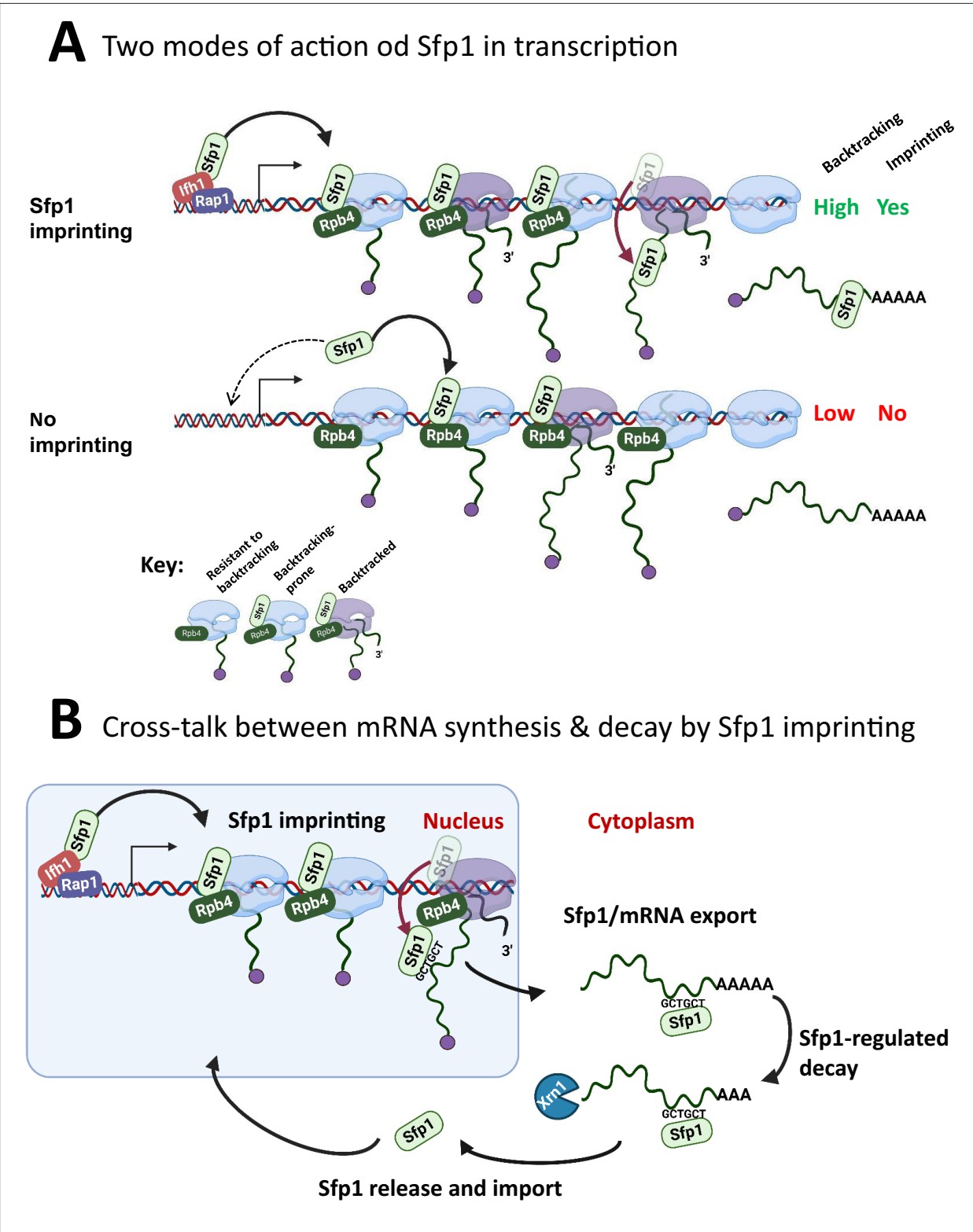

**Figure 8.** A model for split-finger protein 1 (Sfp1) functions in yeast. (**A**) *Two modes of action of Sfp1 in transcription: CRAC + genes recruit Sfp1 to their promoters, whereas non-CRAC + genes recruit Sfp1 from the nuclear space directly to RNA polymerase II (Pol II).* Upper panel represents CRAC + genes that recruit Sfp1 to their promoters. We discovered that Sfp1 appears to accompany Pol II of CRAC + genes, in a manner proportional to the number of transcriptionally active Pol II (*Figure 5A–B*). Its binding to all Pol II molecules, including backtracked Pol II, is even more apparent (*Figure 5B*). In addition, we found that CRAC + genes are enriched with Rap1-binding sites (*Figure 3*). We, therefore, propose that, following binding

*Figure 8 continued on next page*

*Figure 8 continued*

to Rap1-containing promoters, Sfp1 binds Pol II. Specifically, it binds to Rpb4 (and possibly other Pol II subunits) and accompanies it until imprinting. This interaction influences Pol II configuration (*Figure 7A–D*) and increases the likelihood of Pol II to undergo backtracking (*Figure 6B*). Lower panel represents non-CRAC + genes that also interact with Sfp1 (*Figure 5A* CONTROL, *Figure 5B*). We propose that promoters of non-CRAC +genes recruit Sfp1 poorly (relative to CRAC + promoters), except for small group of promoters, e.g., of RiBi genes lacking Rap1 binding site (RapBS). The dashed arrow represents this minor group. For the majority of these genes, the nuclear Sfp1 interacts directly with their elongating Pol II, as its interaction correlates with the extent of chromatin-bound Pol II (*Figure 5B*). This weak interaction also changes Pol II configuration (*Figure 7A*, 'non-CRAC+') and increases the propensity of Pol II to backtrack (*Figure 7B*). However, it either does not result in imprinting or results in rare imprinting events that went undetected by our cross-linking and analysis of cDNA (CRAC) assay. (**B**) Sfp1 mediates cross-talk between mRNA synthesis and decay via imprinting. The backtracked configuration, induced by Sfp1 (**A**), is compatible with a movement of Sfp1 from Pol II to its transcripts (see text), which is enhanced in case the GCTGCT motif is localized near Sfp1 (*Figure 5C*). Following co-transcriptional RNA binding, Sfp1 accompanies the mRNA to the cytoplasm and stabilizes the mRNAs. Following mRNA degradation, Sfp1 is imported back into the nucleus to initiate a new cycle. The model proposes that the specificity of Sfp1-RNA interaction is determined, in part, by the promoter (*Figure 3A–B*). Nevertheless, promoter binding is necessary, but not sufficient for RNA binding. See text for more details.

that the effect of Sfp1 on backtracking is through changing Rpb4 stoichiometry within Pol II or Rpb4 conformation. Backtracking is a conserved process across evolution that influences Pol II processivity and transcription rate (*Bar-Nahum et al., 2005*). We propose an additional potential function: the backtracking-prone configuration of Pol II might be more compatible with mRNA imprinting than the 'regular' configuration (*Figure 8A*). This hypothesis is inspired by the following observations: (I) Sfp1 shows a positive general influence on mRNA stability that is maximal in those mRNAs that it physically binds (*Figure 4A* 'HL,' 4B, and 6 C). (II) Sfp1 physical interaction with mRNAs occurs cotranscriptional (*Figure 5C* and *Figure 2—figure supplement 1D*). (III) The stronger the effect of Sfp1 on Pol II backtracking the stronger its effect on mRNA HL (*Figure 6D*). (IV) Backtracking is substantially more frequent in those genes whose mRNAs are bound by Sfp1 compared to the genome average, and significantly more dependent on this factor (*Figure 6B and C*). Taking all these observations together, it appears that backtracking is associated with co-transcriptional binding of Sfp1 to nascent RNA. Reassuringly, RiBi genes that exhibit Sfp1-dependent backtracking also imprint Sfp1, whereas those that do not also do not imprint (*Figure 6B*, 'RiBi CRAC+,' and 'RiBi non-CRAC+').

Unexpectedly, in addition to its chromatin binding feature, we find that Sfp1 binds a subpopulation of mRNAs whose transcription is stimulated by Sfp1 – 'CRAC +mRNAs.' CRAC + mRNAs exhibit distinctive characteristics compared to other Sfp1-regulated genes (whose mRNAs do not bind Sfp1), or other non-Sfp1-regulated genes (*Figures 2A-C, 3C, 4A, 4B, 5A-C, 6B-D, 7A-B*, *Figure 2—figure supplement 1D, F*, *Figure 4—figure supplement 2A-B*, *Figure 7—figure supplement 1A-B*). Collectively, these findings demonstrate that CRAC + genes and their transcripts constitute a unique group, distinguishable from others based on several criteria, only one of them is the ability of their transcripts to bind Sfp1. CRAC + genes display a broad range of transcription rates (*Figure 6A*) and behave differently from the most highly transcribed genes (*Figure 7A*). Furthermore, while all RiBi mRNAs exhibit similar expression levels, only a subset of them binds to Sfp1. These observations collectively suggest that the 262 CRAC + constitute a distinct set of genes that unveils a new paradigm: genes controlled by a common factor both transcriptionally and post-transcriptionally, via mRNA imprinting.

We observed that RapBS-mediated promoter binding is critical for Sfp1 capacity to bind RNA (*Figure 3*). In RiBi genes, we found highly significant correspondence between the propensity of RiBi promoters to carry RapBS and binding of RiBi mRNAs to Sfp1 (*Figure 3C*). We propose that RNA-binding occurs co-transcriptionally for the following reasons: (I) Sfp1 export from the nucleus to the cytoplasm is dependent on transcription (*Figure 1A*), suggesting that it is exported together with the Pol II transcripts. (II) Splicing occurs co-transcriptionally (e.g. *Churchman and Weissman, 2011*). Consistent with co-transcriptional binding, Sfp1 binds intron-containing *RPL30* RNA (*Figure 2D*; note that the ratio between intron-containing RNA and mature RNA is higher in the IPed lane than in the input one). (III) Binding of Sfp1 to mRNA is dependent on RapBS, suggesting that the same Sfp1 recruited to the promoter by Rap1 binds the transcript. (IV) The ChIP-exo signal drops past the GCTGCT motif in C1-C2 genes (for clusters' definition see *Figure 2B*), a position where Sfp1 prefers to bind the motif containing CRAC + transcripts (*Figure 5C*). These observations suggest that, at least in some of their mRNA targets, Sfp1 is released from Rpb4-containing Pol II to the nascent transcripts co-transcriptionally as GCTGCT motif emerges from Pol II. However, many CRAC + do not contain detectable GCTGCT motifs or have it in a different position (*Figure 2B* C3 cluster). Our observation

that RapBS is required to promote binding to the *RPL30* mRNA sequence, a gene that contains the motif, (*Figure 3B*) demonstrates that motif 1 is not sufficient to recruit Sfp1. Perhaps this motif is used to stabilize the interaction. In the absence of the motif, the movement from Rpb4-containing Pol II to the emerging transcript might be related to the process of polyadenylation (see *Figure 2A*). Collectively, our results unveil a role for CRAC + promoters as mediators between RPB and its interacting RNAs. Whether Rpb4 dissociates together with these proteins remains to be determined. Whether the probability of mRNA imprinting is affected by the time Pol II is engaged in backtracking, until its resolution by TFIIS, is another appealing hypothesis that remains to be examined.

Binding of Sfp1 to mRNAs regulates deadenylation-mediated mRNA decay, mainly by slowing down these processes (*Figure 4*). Following mRNA decay, Sfp1 is imported back to the nucleus (*Figure 1—figure supplement 1B, C*). Following the import, Sfp1 binds to specific promoters and regulates transcription, closing the circle of gene expression regulation (*Figure 8B*). In this way, the synthesis and decay of Sfp1-regulated mRNAs are coordinated in a manner that maintains proper mRNA levels of a specific subset of genes.

The capacity of Sfp1 to bind mature mRNA adds additional complexity to the expression of CRAC + genes. Here, we report that Sfp1 regulates the levels of these gene products by two mechanisms: by stimulation of synthesis and by repression of decay. This raises a possible new mode of regulating mRNA level that targets the Sfp1 imprinting machinery, the extent of which would determine the mRNA HL and consequently the mRNA level. Since Sfp1 enhances both mRNA synthesis and stability, it can serve as a signaling pathway target to rapidly regulate the expression of its clients. Indeed, the expression of Sfp1 targets is highly responsive to the environment (*Albert et al., 2019*; *Jorgensen et al., 2004*).

Sfp1 was viewed as a classical TF that specifically modulates the transcription of a subset of a few hundred genes (see Introduction). We were, therefore, surprised to discover a genome-wide effects of Sfp1 depletion or deletion (*Figures 4B, 6A, D and 7B–C*). These results, in combination with the weak interaction of Sfp1 with chromatin (*Figure 5A* 'Sfp1 Control'), are in accord with previous observation that expression of >30% of the genes, most are not considered to be direct targets of Sfp1, is affected by either Sfp1 depletion or its overexpression (*Albert et al., 2019*, and our unpublished NET-seq data). An important support for this notion is the correlation we found between the binding of Sfp1 to gene bodies and Pol II occupancy in all genes (*Figure 5B*). The general capacity of Sfp1 to alter elongating Pol II in the context of Rpb4 and to affect backtracking (*Figure 7B*, *Figure 7—figure supplement 1A*) likely underlies its widespread effect on transcription.

As mentioned above, this widespread effect of Sfp1 on backtracking correlates with higher mRNA HL in many genes (not only in CRAC+; *Figure 6D*). This suggests that the effect of Sfp1-dependent backtracking on mRNA HL does not always require Sfp1 binding to mRNA. Perhaps, this indirect effect of Sfp1 on mRNA HL is mediated by other factors, like Ccr4-Not, whose imprinting may also be affected by Pol II backtracking (*Kruk et al., 2011*; *Villanyi et al., 2014*; *Begley et al., 2019*). In contrast, the correlation between Rpb4/Rpb3 and mRNA HL is much higher in CRAC + than in the rest of the genome, and is entirely dependent on Sfp1 (*Figure 7C*, *Figure 7—figure supplement 1B*). Interestingly, we previously demonstrated that transcription elongation of Rap1-controlled genes is exceptionally affected by backtracking (*Pelechano et al., 2009*). We propose that Rap1-dependent recruitment of Sfp1 modulates its interaction with Pol II in a manner that permits Sfp1 transfer to mRNA (*Figure 8*).

Binding of Sfp1 to CRAC + mRNAs, which stabilizes the bound mRNAs, is dependent on Rap1 binding sites. On the other hand, we previously found that depletion of Rap1 leads to mRNA stabilization (*Bregman et al., 2011*). If the effect of Rap1 on mRNA stability is mediated only through Sfp1, we would expect that its depletion should result in mRNA destabilization, contrary to our previous observation (*Bregman et al., 2011*). Therefore, we conclude that Rap1 mediates additional activity that enhances mRNA decay and overrides the destabilizing effect of Sfp1, which remains to be discovered.

The Rpb4 stoichiometry within yeast Pol II is <1 (*Choder and Young, 1993*). It was not clear whether the sub-stoichiometric feature of Rpb4 is constant across all Pol II molecules. Here, we show that stoichiometry, as reflected by the Rpb4 ChIP/Rpb3 ChIP ratio, decreases gradually during transcription elongation. It is possible that there is a link between the Rpb4 capacity to modulate backtracking and the gradual decrease in its stoichiometry. Interestingly Sfp1 affects this stoichiometry drop, thus serving as a regulator of Rpb4 stoichiometry. Moreover, the effect of Sfp1 on mRNA stability is entirely

linked to its action on Rpb4 stoichiometry (*Figure 7C* and *Figure 7—figure supplement 1B*). We note, however, that, although changes in the stoichiometry are the most likely interpretation of the change in Rpb4 ChIP/Rpb3 ChIP ratio, this change can merely reflect changes in Pol II configuration that compromise the capacity of Rpb4 to produce ChIP signal.

Interestingly, the Sfp1-mediated mechanism contrasts with that of mRNA buffering. The known model of mRNA buffering posits that a modulation of one process (e.g. transcription) is balanced by a reciprocal modulation of the other process (e.g. mRNA decay), thus maintaining the mRNA level constant, or nearly constant (*Bryll and Peterson, 2023*; *Haimovich et al., 2013b*; *Pérez-Ortín and Chávez, 2022*; *Timmers and Tora, 2018*). In contrast, the functions of Sfp1 do not result in balancing, but the contrary: the two activities of Sfp1 cooperatively increase mRNA level – increase mRNA synthesis and decrease its degradation. Thus, Sfp1 is a paramount example of enhanced gene regulation by cooperation between mRNA decay and gene transcription that we previously modeled (*García-Martínez et al., 2023*). Moreover, we reported that mRNA buffering maintains a constant mRNA concentration regardless of the strain growth rate, except for growth-related genes (*García-Martínez et al., 2016*; *Chattopadhyay et al., 2022*; *Pérez-Ortín and Chávez, 2022*). The expression of the latter genes increases with growth rate. The discovered effect of Sfp1 on its target genes, most of which are growth-related, provides a plausible mechanism to explain how growth-related genes avoid buffering when their expression must be rapidly adjusted to the ever-changing environment.

In summary, we propose that the role of certain class-specific TFs, such as Sfp1, extends beyond merely controlling various stages of mRNA synthesis and processing in the nucleus; they also regulate post-transcriptional functions in the cytoplasm. We hypothesize that a single factor (or a complex of factors) has evolved the capacity to regulate both transcriptional and post-transcriptional functions to facilitate cross-talk between these two mechanisms. Sfp1 impacts the mRNA life cycle - from transcription to degradation. Interestingly, the Sfp1 effect on transcription elongation is linked to its function on mRNA stabilization, suggesting that all its functions are connected. Sfp1 possesses two classical zinc fingers. Given that proteins with zinc fingers, characteristic of many TFs, are known to be involved in DNA, RNA, and protein binding, it is conceivable that this zinc finger-containing protein is suitable for participating in the mRNA imprinting process, as proposed here (*Figure 8B*). Given that RBPs have been proposed to perform multiple roles in RNA-based regulation of gene expression in mammals (*Xiao et al., 2019*), we anticipate that the case of Sfp1 as an imprinting factor will serve an example of this important type of gene regulators in other eukaryotes.

## Materials and methods
### Yeast strains and plasmids construction
Yeast strains, and plasmids are listed in *Supplementary file 1*. HISx6-TEV protease site-Protein A (HTP) tag was inserted into the chromosome by homologous recombination with PCR amplified fragment carrying *SFP1p::CaURA3::SFP1p*:: HTP::*SFP1* ORF (only 100 bp repeats of the promoter and 100 bp of the ORF). After integration into the *SFP1* locus, the *Ca::URA3* was popped out by selection on 5-FOA, utilizing the two identical *SFP1p* repeats, recreating the 5' non-coding region. In this way, the tag was surgically introduced without adding other sequences. All strains were verified by both PCR and Sanger sequencing.

### Yeast proliferation conditions, under normal and fluctuating temperatures, and during exit from starvation
Yeast cells were grown in synthetic complete (SC), Synthetic dropout (SD), or in YPD medium at 30 °C unless otherwise indicated; for harvesting optimally proliferating cells, strains were grown for at least seven generations in the logarithmic phase before harvesting.

### Genomic run-on and degron procedures
Genomic run-on (GRO) was performed as described in *García-Martínez et al., 2004*, as modified in *Oliete-Calvo et al., 2018*. Briefly, GRO detects by macroarray hybridization, genome-wide, active elongating RNA pol II, whose density per gene is taken as a measurement of its SR. Total SR for a given yeast strain was calculated as the sum of all individual gene SRs. At the same time, the protocol allows the mRNA amounts (RA) for all the genes to be measured by means of the hybridization of

labeled cDNA onto the same nylon filters. Total mRNA concentration in yeast cells was determined by quantifying polyA + in total RNA samples by oligo-dT hybridization of a dot-blot following the protocol described in *García-Martínez et al., 2004*. mRNA half-lives, in arbitrary units, are calculated as RA/SR by assuming steady-state conditions for the transcriptome. All the experiments were done in triplicate.

For the degron experiment, the auxin degron strain (AID-Sfp1) was inoculated in YPD medium until an $OD_{600}$ of 0.4 and then an aliquot was taken, corresponding to t=0. After auxin addition two more samples were taken at 20 and 60 min. At each time point (0, 20, and 60 min) Genomic Run-on (GRO) was performed, as described in the previous paragraph, and for SR, RA, and HL data were obtained for each time point.

GEO accession numbers for the genomic data are: GSE57467 and GSE202748.

## Performing and analyzing the UVCRAC

The endogenous Sfp1 gene was surgically tagged with an N-terminal ProteinA-TEV-His6 (PTH)tag, retaining intact 5' and 3' non-coding regions, as described above. Two WT-independent colonies and two *rpb4Δ* strains (yMC1019-1022) were subject to CRAC, utilizing a previously reported protocol (*Haag et al., 2017*). Briefly, yeast cells were allowed to proliferate in a synthetic complete medium at 30 °C till $1×10^7$ cells/ml, UV irradiated, and harvested and frozen in liquid nitrogen. Frozen cells were pulverized cryogenically using a mixer mill 400 (RETSCH). PTH-Sfp1 was isolated, RNA fragments cross-linked to Sfp1 was end-labeled and purified, converted to a cDNA library, and sequenced by next-generation sequencing NGS 500 (all sequence data are available from GEO under accession number **GSE230761**).

The CRAC sequencing results were processed by the CRAC pipeline, using the pyCRAC package (PMCID: PMC4053934) for single-end reads that the Granneman lab previously developed (*van Nues et al., 2017*). Briefly, low-quality sequences and adapter sequences were trimmed using Flexbar (version 3.5.0; PMID: 24832523). Reads were then collapsed using random barcode information provided in the in-read barcodes using pyFastqDupicateRemover.py from the pyCRAC package (PMCID: PMC4053934). Collapsed reads were aligned to the yeast reference genome (R64) using novoalign (https://www.novocraft.com/) version 2.0.7. Reads that mapped to multiple genomic regions were randomly distributed over each possible location. PyReadCounters.py was then used, to make read count and fragments per kilobase transcript per million reads (FPKM) tables for each annotated genomic feature.

## Bioinformatics analysis of CRAC, ChIP-exo, and ChEC-seq datasets

The CRAC sequencing results were processed by a pipeline that we developed previously (*van Nues et al., 2017*). High-quality reads were mapped against the *Saccharomyces cerevisiae* sacCer3 (R64) reference genome. ChIP-exo (*Reja et al., 2015*) and ChEC-seq (*Albert et al., 2019*) raw sequencing datasets were retrieved from accessions PRJNA245761 and PRJNA486090, respectively. All fastq files were also aligned to the *Saccharomyces cerevisiae* R64 reference genome with Bowtie2, using default parameters. Alignment files were used to generate average metagene profiles with the ngs.plot R package (Shen, L., et al. 2014). The robust parameter -RB was used in every plot to filter genes with the 5% most extreme coverage values. Ngs.plot was also used to generate CRAC signal heatmaps around pA sites, and the -GO km function was used to perform k-means clustering of genes with similar profiles. Venn diagrams were generated in R, and the overlap was statistically evaluated for overrepresentation or depletion using the hypergeometric test. For motif analysis, 160 bp sequences around pA sites were extracted in fasta format from the 113 CRAC + genes in clusters 1 and 2 (*Figure 2B*) with the sequence tools function from the RSAT website (http://rsat.france-bioinformatique.fr/fungi/) and used with two sets of motif discovery tools: MEME (https://meme-suite.org/meme/tools/meme), and DRImust (http://drimust.technion.ac.il). For MEME we used the differential enrichment mode, establishing a control set of 262 sequences belonging to non-CRAC + genes randomly selected but excluding RP and RiBi. We set up a minimum motif size of 6 nt, and a maximum of 50 nt. We also restricted the search to the same strand of the gene analyzed. For DRImust, we also used a target vs background sequences strategy, a strand-specific search mode, and the same minimum and maximum motif lengths as MEME. Both tools encountered the same GCTGCT consensus motif, but in overlapping but slightly different sets of genes. We fused the lists of genes from both tools for downstream

analysis. GO term enrichment analysis (biological process ontology) was done with the enrichment tool from Yeastmine (https://yeastmine.yeastgenome.org/yeastmine/begin.do), selecting only GO terms with Holm-Bonferroni-corrected p-values<0.05. Spearman correlation values and statistical test result asterisks were inserted into ggplot2-generated plots by using the package ggpubr (https://rpkgs.datanovia.com/ggpubr).

## ChIP on chip experiments

Chromatin immunoprecipitation was performed as previously described (*Rodríguez-Gil et al., 2010*) using ab81859 (Abcam) anti-Rpb3 and 9E10 (Santa Cruz Biotechnology) anti-C-Myc antibodies. After crosslinking reversal, the obtained fragments (300 bp approximately) of enriched DNA were amplified unspecifically and labelled following Affymetrix Chromatin Immunoprecipitation Assay Protocol P/N 702238. Genomic DNA controls were processed in parallel. 10 μl of each sample were amplified using Sequenase. The reaction mix consisted of 10 μl purified DNA, 4 μl 5 X Sequenase reaction buffer, and 4 μl Primer A (200 μM) for each reaction. The cycle conditions for random priming were 95 °C for 4 min, snap cool on ice, and hold at 10 °C. Next, 2.6 μl of 'first cocktail' (0.1 μl 20 mg mg/ml BSA), 1 μl 0.1 M DTT, 0.5 μl 25 mM dNTPs, and 1 μl diluted Sequenase 1/10 from 13 U/μl stock were added to each reaction and put back in the thermocycler for the following program: 10 °C for 5 min, ramp from 10–37°C over 9 min, 37 °C for 8 min, 95 °C for 4 min, snap cool on ice and 10 °C hold. Then another 1 μl of cocktail was added to each sample and these steps were repeated for two more cycles. The samples were kept at a 4 °C hold. After PCR amplification with dUTP, the samples were purified using Qiagen QIAquick PCR Purification Kit (50) (Cat.No. 28104). About 56 μl of first-round purified DNA were collected for each reaction. The amplification PCR was performed as usual but using 20 μl of first-round DNA from the previous step, 3.75 μl of a 10 mM dNTPs + dUTP mix and 4 μl of 100 μM Primer B. The cycling conditions were: 15 cycles consisting of 95 °C for 30 s, 45 °C for 30 s, 55 °C for 30 s and 72 °C for 1 min, and 15 cycles of 95 °C for 30 , 45 °C for 30 s, 55 °C for 30 s, and 72 °C for 1 min, adding 5 s for every subsequent cycle. DNA quality and quantity were checked in a 1% agarose gel and using a NanoDrop ND-1000 Spectrophotometer. The samples were purified using the QIAquick PCR purification kit (Qiagen). Then 0.5 μg of each was used to hybridize GeneChip *S. cerevisiae* Tiling 1.0 R custom arrays. This step was carried out in the Multigenic Analysis Service of the University of Valencia. The obtained CEL archives were normalized by quantile normalization and the intensities of the signal were extracted using the TAS (Tiling Analysis Software) developed by Affymetrix. The resulting text files were read using R scripts to adjudicate probe intensities to genes. The $log_2$ values of the median intensities of the chosen different groups of genes were represented. In order to compare the data between different experiments the values were normalized by median and standard deviation (z-score $log_2$).

## mRNA decay assay

The assay was performed as described (*Lotan et al., 2005*). Briefly, optimally proliferating cells were treated with thiolutin, or shifted to 42 °C, to stop transcription, and cell samples were harvested at various time points post-transcription arrest. RNA was extracted and equal amounts of total RNA were loaded on an agarose gel for Northern blot hybridization, using radiolabeled DNA probes. The same membrane was sequentially hybridized with the indicated probes. SCR1 RNA (Pol III transcript) and rRNAs were used for normalization. For quantification: The mRNAs levels were determined by PhosphoImager technology. mRNA level was normalized to that of the Pol III transcript *SCR1*. Band intensity at time 0, before transcription inhibition, was defined as 100% of the initial mRNA level and the intensities at the other time points were calculated relative to time 0.

## Polyacrylamide northern (PAGEN) analysis

PAGEN was performed as described previously (*Lotan et al., 2005*; *Sachs and Davis, 1989*). Briefly, equal amounts of total RNA pellets were suspended in ~15 μl formamide loading dye and loaded on a 20 × 20 × 0.1 cm gel of 6% polyacrylamide, 7 M urea in 1X TBE (tris borate EDTA) buffer. Following electrophoresis, the gel was then subjected to electro-transfer onto a nylon filter (GeneScreen Plus) in 0.5X TBE at 30 Volts for 7–15 hr at 4 °C. Following blotting, the PAGEN filter was reacted with radioactive probes, as described (*Lotan et al., 2005*).

## Acknowledgements

We thank David Shore for the gift of the AID-SFP1 strain and for critically reading the manuscript. This work was supported by the Israel Science Foundation grant # 301/20 (to MC) and by grants [PID2020-112853GB-C31 funded by MCIN/AEI/10.13039/501100011033 to JEP-O and grant PID2020-112853GB-C32 funded by MCIN/AEI/10.13039/501100011033 to SC]. SG was supported by a Medical Research Council Non-Clinical Senior Research Fellowship [MR/R008205/1 to S.G.]

## Additional information

### Funding

| Funder | Grant reference number | Author |
|---|---|---|
| Israel Science Foundation | 301/20 | Mordechai Choder |
| MCIN/ AEI/10.13039/501100011033 | PID2020-112853GB-C31 | José E Pérez-Ortín |
| MCIN/ AEI/10.13039/501100011033 | PID2020-12853GB-C32 | Sebastián Chávez |
| Medical Research Council Non-Clinical Senior Research Fellowship | MR/R008205/1 to S.G. | Sander Granneman |

The funders had no role in study design, data collection and interpretation, or the decision to submit the work for publication.

### Author contributions

Moran Kelbert, Data curation, Investigation, Methodology; Antonio Jordán-Pla, Conceptualization, Formal analysis, Investigation; Lola de Miguel-Jiménez, Data curation, Investigation; José García-Martínez, Software, Formal analysis, Methodology; Michael Selitrennik, Adi Guterman, Noa Henig, Investigation; Sander Granneman, Data curation, Software, Formal analysis; José E Pérez-Ortín, Conceptualization, Supervision, Funding acquisition, Investigation, Methodology, Writing – review and editing; Sebastián Chávez, Conceptualization, Supervision, Funding acquisition, Writing – review and editing; Mordechai Choder, Conceptualization, Supervision, Funding acquisition, Writing - original draft, Writing – review and editing

### Author ORCIDs

José E Pérez-Ortín  https://orcid.org/0000-0002-1992-513X
Sebastián Chávez  https://orcid.org/0000-0002-8064-4839
Mordechai Choder  https://orcid.org/0000-0003-0187-2110

Reviewer #1 (Public Review): https://doi.org/10.7554/eLife.90766.4.sa1
Author response https://doi.org/10.7554/eLife.90766.4.sa2

## Additional files

### Supplementary files

- MDAR checklist
- Supplementary file 1. Yeast strains used in this study.
- Supplementary file 2. Yeast strains used in this study.

### Data availability

All data generated or analysed during this study are included in the manuscript and supporting files.

The following datasets were generated:

| Author(s) | Year | Dataset title | Dataset URL | Database and Identifier |
|---|---|---|---|---|
| Pérez-Ortín JE, García-Martínez J | 2023 | Transcriptomic analysis of the effect of Sfp1 depletion | https://www.ncbi.nlm.nih.gov/geo/query/acc.cgi?acc=GSE202748 | NCBI Gene Expression Omnibus, GSE202748 |
| Kelbert M, Choder M, Pérez-Ortín JE, Chávez S, Jordán-Pla A, Granneman S | 2023 | The transcription factor Sfp1 imprints specific classes of mRNAs and links their synthesis and cytoplasmic decay | https://www.ncbi.nlm.nih.gov/geo/query/acc.cgi?acc=GSE230761 | NCBI Gene Expression Omnibus, GSE230761 |

The following previously published dataset was used:

| Author(s) | Year | Dataset title | Dataset URL | Database and Identifier |
|---|---|---|---|---|
| García-Martínez J, Pérez-Ortín JE, Medina DA, Choder M, Chávez S | 2015 | Genomic Run On (GRO): determination of the nascent transcriptional rates and mRNA levels in several yeast mutants | https://www.ncbi.nlm.nih.gov/geo/query/acc.cgi?acc=GSE57467 | NCBI Gene Expression Omnibus, GSE57467 |

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
