## [Editor Report · eLife assessment]

This **important** study reports that a transcription factor stimulating mRNA synthesis can stabilize its target transcripts. The **convincing** results demonstrate, with multiple independent approaches, co-transcriptional binding, stabilization of a family of mRNAs, and cytoplasmic activities of the transcription factor Sfp1. The results lead to the conclusion that the co-transcriptional association of Sfp1 with specific transcripts is a critical step in the stabilization of such transcripts in the cytoplasm.

---

## [Referee Report · Reviewer #1 (Public Review)]

This manuscript builds upon the authors' previous work on the cross-talk between transcription initiation and post-transcriptional events in yeast gene expression. These prior studies identified an mRNA 'imprinting' phenomenon linked to genes activated by the Rap1 transcription factor (TF), a surprising role for the Sfp1 TF in promoting RNA polymerase II (RNAPII) backtracking, and a role for the non-essential RNAPII subunits Rpb4/7 in the regulation of mRNA decay and translation. Here the authors aimed to extend these observations to provide a more coherent picture of the role of Sfp1 in transcription initiation and subsequent steps in gene expression. They provide evidence for (1) a physical interaction between Sfp1 and Rpb4, (2) Sfp1 binding and stabilization of mRNAs derived from genes whose promoters are bound by both Rap1 and Sfp1 and (3) an effect of Sfp1 on Rpb4 binding or conformation during transcription elongation.

---

## [Author Response]

The following is the authors’ response to the previous reviews.

**Public Reviews:**

**Reviewer #2 (Public Review):**
Summary:The manuscript by Kelbert et al. presents results on the involvement of the yeast transcription factor Sfp1 in the stabilisation of transcripts whose synthesis it stimulates. Sfp1 is known to affect the synthesis of a number of important cellular transcripts, such as many of those that code for ribosomal proteins. The hypothesis that a transcription factor can remain bound to the nascent transcript and affect its cytoplasmic half-life is attractive. However, the association of Sfp1 with cytoplasmic transcripts remains to be validated, as explained in the following comments:A two-hybrid based assay for protein-protein interactions identified Sfp1, a transcription factor known for its effects on ribosomal protein gene expression, as interacting with Rpb4, a subunit of RNA polymerase II. Classical two-hybrid experiments depend on the presence of the tested proteins in the nucleus of yeast cells, suggesting that the observed interaction occurs in the nucleus. Unfortunately, the two-hybrid method cannot determine whether the interaction is direct or mediated by nucleic acids. The revised version of the manuscript now states that the observed interaction could be indirect.To understand to which RNA Sfp1 might bind, the authors used an N-terminally tagged fusion protein in a cross-linking and purification experiment. This method identified 264 transcripts for which the CRAC signal was considered positive and which mostly correspond to abundant mRNAs, including 74 ribosomal protein mRNAs or metabolic enzyme-abundant mRNAs such as PGK1. The authors did not provide evidence for the specificity of the observed CRAC signal, in particular what would be the background of a similar experiment performed without UV cross-linking. This is crucial, as Figure S2G shows very localized and sharp peaks for the CRAC signal, often associated with over-amplification of weak signal during sequencing library preparation.

(1) To rule out possible PCR artifacts, we used a UMI (Unique Molecular Identifier) scan. UMIs are short, random sequences added to each molecule by the 5’ adapter to uniquely tag them. After PCR amplification and alignment to the reference genome, groups of reads with identical UMIs represent only one unique original molecule. Thus, UMIs allow distinguishing between original molecules and PCR duplicates, effectively eliminating the duplicates.

(2) Looking closely at the peaks using the IGV browser, we noticed that the reads are by no means identical. Each carrying a mutation [probably due to the cross-linking] in a different position and having different length. Note that the reads are highly reproducible in two replicate.

(3) CRAC+ genes do not all fall into the category of highly transcribed genes. On the contrary, as depicted in Figure 6A (green dots), it is evident that CRAC+ genes exhibit a diverse range of Rpb3 ChIP and GRO signals. Furthermore, as illustrated in Figure 7A, when comparing CRAC+ to Q1 (the most highly transcribed genes), it becomes evident that the Rpb4/Rpb3 profile of CRAC+ genes is not a result of high transcription levels.

(4) Only a portion of the RiBi mRNAs binds Sfp1, despite similar expression of all RiBi.

(5) The CRAC+ genes represent a distinct group with many unique features. Moreover, many CRAC+ genes do not fall into the category of highly transcribed genes.

(6) The biological significance of the 262 CRAC+ mRNAs was demonstrated by various experiments; all are inconsistent with technical flaws. Some examples are:

a) Fig. 2a and B show that most reads of CRAC+ mRNA were mapped to specific location – close the pA sites.

b) Fig. 2C shows that most reads of CRAC+ mRNA were mapped to specific RNA motif.

c) Most RiBi CRAC+ promoter contain Rap1 binding sites (p = 1.9x10-22), whereas the vast majority of RiBi CRAC- promoters do not contain Rap1 binding site. (Fig. 3C).

d) Fig. 4A shows that RiBi CRAC+ mRNAs become destabilized due to Sfp1 deletion, whereas RiBi CRAC- mRNAs do not. Fig. 4B shows similar results due to

e) Fig. 6B shows that the impact of Sfp1 on backtracking is substantially higher for CRAC+ than for CRAC- genes. This is most clearly visible in RiBi genes.

f) Fig. 7A shows that the Sfp1-dependent changes along the transcription units is substantially more rigorous for CRAC+ than for CRAC-.

g) Fig. S4B Shows that chromatin binding profile of Sfp1 is different for CRAC+ and CRAC- genes

In a validation experiment, the presence of several mRNAs in a purified SFP1 fraction was measured at levels that reflect the relative levels of RNA in a total RNA extract. Negative controls showing that abundant mRNAs not found in the CRAC experiment were clearly depleted from the purified fraction with Sfp1 would be crucial to assess the specificity of the observed protein-RNA interactions (to complement Fig. 2D).

GPP1, a highly expressed genes, is not to be pulled down by Sfp1 (Fig. 2D). GPP1 (alias RHR2) was included in our Table S2 as one of the 264 CRAC+ genes, having a low CRAC value. However, when we inspected GPP1 results using the IGV browser, we realized that the few reads mapped to GPP1 are actually anti-sense to GPP1 (perhaps they belong to the neighboring RPL34B genes, which is convergently transcribed to GPP1) (see Fig. 1 at the bottom of the document). Thus, GPP1 is not a CRAC+ gene and would now serve as a control. See We changed the text accordingly (see page 11 blue sentences). In light of this observation, we checked other CRAC genes and found that, except for ALG2, they all contain sense reads (some contain both sense and anti-sense reads). ALG2 and GPP1 were removed leaving 262 CRAC+ genes.

The CRAC-selected mRNAs were enriched for genes whose expression was previously shown to be upregulated upon Sfp1 overexpression (Albert et al., 2019). The presence of unspliced RPL30 pre-mRNA in the Sfp1 purification was interpreted as a sign of co-transcriptional assembly of Sfp1 into mRNA, but in the absence of valid negative controls, this hypothesis would require further experimental validation. Also, whether the fraction of mRNA bound by Sfp1 is nuclear or cytoplasmic is unclear.

Further experimental validation was provided in some of our figures (e.g., Fig. 5C, Fig. 3B).

We argue that Sfp1 binds RNA co-transcriptionally and accompanies the mRNA till its demise in the cytoplasm: Co-transcriptional binding is shown in: (I) a drop in the Sfp1 ChIP-exo signal that coincides with the position of Sfp1 binding site in the RNA (Fig. 5C), demonstrating a movement of Sfp1 from chromatin to the transcript, (II) the dependence of Sfp1 RNA-binding on the promoter (Fig. 3B) and binding of intron-containing RNA. Taken together these 3 different experiments demonstrate that Sfp1 binds Pol II transcript co-transcriptionally. Association of Sfp1 with cytoplasmic mRNAs is shown in the following experiments: (I) Figure 2D shows that Sfp1 pulled down full length RNA, strongly suggesting that these RNA are mature cytoplasmic mRNAs. (II) mRNA encoding ribosomal proteins, which belong to the CRAC+ mRNAs group are degraded by Xrn1 in the cytoplasm (Bresson et al., Mol Cell 2020). The capacity of Sfp1 to regulates this process (Fig. 4A-D) is therefore consistent with cytoplasmic activity of Sfp1. (III) The effect of Sfp1 on deadenylation (Fig. 4D), a cytoplasmic process, is also consistent with cytoplasmic activity of Sfp1.

To address the important question of whether co-transcriptional assembly of Spf1 with transcripts could alter their stability, the authors first used a reporter system in which the RPL30 transcription unit is transferred to vectors under different transcriptional contexts, as previously described by the Choder laboratory (Bregman et al. 2011). While RPL30 expressed under an ACT1 promoter was barely detectable, the highest levels of RNA were observed in the context of the native upstream RPL30 sequence when Rap1 binding sites were also present. Sfp1 showed better association with reporter mRNAs containing Rap1 binding sites in the promoter region. Removal of the Rap1 binding sites from the reporter vector also led to a drastic decrease in reporter mRNA levels. Co-purification of reporter RNA with Sfp1 was only observed when Rap1 binding sites were included in the reporter. Negative controls for all the purification experiments might be useful.

In the swapping experiment, the plasmid lacking RapBS serves as the control for the one with RapBS and vice versa (see Bregman et al., 2011). Remember, that all these contracts give rise to identical RNA. Indeed, RabBS affects both mRNA synthesis and decay, therefore the controls are not ideal. However, see next section.

More importantly, in Fig. 3B “Input” panel, one can see that the RNA level of “construct F” was higher than the level of “construct E”. Despite this difference, only the RNA encoded by construct E was detected in the IP panel. This clearly shows that the detection of the RNA was not merely a result of its expression level.

To complement the biochemical data presented in the first part of the manuscript, the authors turned to the deletion or rapid depletion of SFP1 and used labelling experiments to assess changes in the rate of synthesis, abundance and decay of mRNAs under these conditions. An important observation was that in the absence of Sfp1, mRNAs encoding ribosomal protein genes not only had a reduced synthesis rate, but also an increased degradation rate. This important observation needs careful validation,

Indeed, we do provide validations in Fig. 4C Fig. 4D Fig. S3A and during the revision we included an additional validation as Fig. S3B. Of note, we strongly suspect that GRO is among the most reliable approaches to determine half-lives (see our response in the first revision letter).

As genomic run-on experiments were used to measure half-lives, and this particular method was found to give results that correlated poorly with other measures of half-life in yeast (e.g. Chappelboim et al., 2022 for a comparison). As an additional validation, a temperature shift to 42{degree sign}C was used to show that , for specific ribosomal protein mRNA, the degradation was faster, assuming that transcription stops at that temperature. It would be important to cite and discuss the work from the Tollervey laboratory showing that a temperature shift to 42{degree sign}C leads to a strong and specific decrease in ribosomal protein mRNA levels, probably through an accelerated RNA degradation (Bresson et al., Mol Cell 2020, e.g. Fig 5E).

This was cited. Thank you.

Finally, the conclusion that mRNA deadenylation rate is altered in the absence of Sfp1, is difficult to assess from the presented results (Fig. 3D).

This type of experiment was popular in the past. The results in the literature are similar to ours (in fact, ours are nicer). Please check the papers cited in our MS and a number of papers by Roy Parker.

The effects of SFP1 on transcription were investigated by chromatin purification with Rpb3, a subunit of RNA polymerase, and the results were compared with synthesis rates determined by genomic run-on experiments. The decrease in polII presence on transcripts in the absence of SFP1 was not accompanied by a marked decrease in transcript output, suggesting an effect of Sfp1 in ensuring robust transcription and avoiding RNA polymerase backtracking. To further investigate the phenotypes associated with the depletion or absence of Sfp1, the authors examined the presence of Rpb4 along transcription units compared to Rpb3. An effect of spf1 deficiency was that this ratio, which decreased from the start of transcription towards the end of transcripts, increased slightly. To what extent this result is important for the main message of the manuscript is unclear.Suggestions: (a) please clearly indicate in the figures when they correspond to reanalyses of published results.

This was done.

(b) In table S2, it would be important to mention what the results represent and what statistics were used for the selection of "positive" hits.

This was discussed in the text.

Strengths:- Diversity of experimental approaches used.- Validation of large-scale results with appropriate reporters.Weaknesses:- Lack of controls for the CRAC results and lack of negative controls for the co-purification experiments that were used to validate specific mRNA targets potentially bound by Sfp1.- Several conclusions are derived from complex correlative analyses that fully depend on the validity of the aforementioned Sfp1-mRNA interactions.

We hope that our responses to Reviewer 2's thoughtful comments have rulled out concerns regarding the lack of controls.

**Recommendations for the authors:**

**Reviewer #2 (Recommendations For The Authors):**
Please review the text for spelling errors. While not mandatory, wig or begraph files for the CRAC results would be very useful for the readers.

**Author response image 1. sa2fig1:** A snapshot of IGV GPP1 locus showing that all the reads are anti-sense pointing at the opposite direction of the gene (the gene arrows [white arrows over blue, at the bottom] are pointing to the right whereas the reads’ orientations are pointing to the left).